# Robust Binary Models by Pruning Randomly-initialized Networks

**Chen Liu**\* **Ziqi Zhao**\* **Sabine Süsstrunk** **Mathieu Salzmann**
École Polytechnique Fédérale de Lausanne, Lausanne, Switzerland,
{chen.liu, ziqi.zhao, sabine.susstrunk, mathieu.salzmann}@epfl.ch

## Abstract

Robustness to adversarial attacks was shown to require a larger model capacity, and thus a larger memory footprint. In this paper, we introduce an approach to obtain robust yet compact models by pruning randomly-initialized binary networks. Unlike adversarial training, which learns the model parameters, we initialize the model parameters as either $+1$ or $-1$, keep them fixed, and find a subnetwork structure that is robust to attacks. Our method confirms the *Strong Lottery Ticket Hypothesis* in the presence of adversarial attacks, and extends this to binary networks. Furthermore, it yields more compact networks with competitive performance than existing works by 1) adaptively pruning different network layers; 2) exploiting an effective binary initialization scheme; 3) incorporating a last batch normalization layer to improve training stability. Our experiments demonstrate that our approach not only always outperforms the state-of-the-art robust binary networks, but also can achieve accuracy better than full-precision ones on some datasets. Finally, we show the structured patterns of our pruned binary networks.

## 1 Introduction

Deep neural networks have achieved unprecedented success in machine learning [19, 30, 56]. However, their state-of-the-art performance comes with costs. First, modern deep neural networks usually have millions of parameters, making them difficult to deploy on devices with limited memory or computational power. Second, these models are vulnerable to adversarial attacks: imperceptible perturbations of the input can dramatically change their output and lead to incorrect predictions [55]. Furthermore, jointly addressing both issues is complicated by the fact that, as shown in [44, 60], achieving robustness against adversarial attacks typically requires higher network capacity.

In this paper, we introduce an approach to obtaining compact and robust binary neural networks. Our method follows a fundamentally different philosophy from typical adversarial training [44]: instead of using adversarial examples to train the model parameters, we fix the model parameters and search for a robust network structure. To this end, and to simultaneously achieve compactness, we prune randomly-initialized binary networks. The resulting sparse and binary networks have much smaller memory footprint than the dense or full-precision ones. They are inherently lightweight and robust.

Our work is motivated by the *Strong Lottery Ticket Hypothesis* [51], which observed that within a random overparameterized network, there exists a subnetwork achieving performance similar to that of trained networks with the same number of parameters. The *Robust Scratch Ticket* [22] extends this hypothesis to the context of adversarial robustness. Here, we introduce a novel pruning strategy that yields higher compression rates, and investigate the case of binary model parameters. Specifically, we develop an *adaptive pruning strategy* to adaptively use different pruning rates for different layers. Furthermore, we introduce a normalization-based technique to increase the algorithm's stability for

---

\*indicates equal contributions

36th Conference on Neural Information Processing Systems (NeurIPS 2022).

binary networks. The subnetworks obtained by our method are consequently more compact than a full-precision one, while achieving a similar robustness to attacks.

We conduct extensive experiments on standard benchmarks to confirm the effectiveness of our method. We obtain both better performance and a more stable training behavior than existing works [22]. Furthermore, our approach outperforms the state-of-the-art robust binary networks [25], achieving performance on par with or even better than the state-of-the-art robust full-precision ones [8, 46, 54] while producing much more compact networks.

Finally, we conduct preliminary investigations on the structure of the robust subnetworks obtained by our algorithm. We find our methods prefer to prune the whole channel or the kernel in the convolutional layers. In addition, for two consecutive convolutional layers, the kernels pruned in the first layer are well aligned with the channels pruned in the second layer. Altogether, our work sheds some light on understanding the structure of robust networks of high parameter sparsity, it also indicates the potential of regular pruning.

Our code is available at https://github.com/IVRL/RobustBinarySubNet.

**Notation.** We use light letters, lowercase bold letters, and uppercase bold letters to represent scalars, vectors, and higher dimensional tensors, respectively. $\odot$ is the elementwise multiplication operation. We use the term *adversarial budget* to represent the range of allowable perturbations. Specifically, the adversarial budget $\mathcal{S}_\epsilon$ is based on the $l_\infty$ norm and defined as $\{\Delta | \|\Delta\|_\infty \leq \epsilon\}$, with $\epsilon$ the strength of the adversarial budget. We refer to the proportion of pruned parameters over the total number of parameters in a layer or a model as the *pruning rate $r$*.

## 2   Related Work

**Adversarial Robustness.** Deep neural networks have been shown to be vulnerable to adversarial attacks [55, 45]. To generate adversarial examples, the *Fast Gradient Sign Method* (FGSM) [24] perturbs the input in the direction of the input gradient. The *Iterative Fast Gradient Sign Method* (IFGSM) [38] improves FGSM by running it iteratively. *Projected Gradient Descent* (PGD) [44] uses random initialization and multiple restarts on top of IFGSM to further strengthen the attack. Recently, AutoAttack (AA) [15] has led to state-of-the-art attacks by ensembling different types of attacks; it is used to reliably benchmark the robustness of models [13] and we thus use it in our experiments.

Many works have proposed defense mechanisms against these adversarial attacks. Early ones [5, 47, 59] used obfuscated gradients [3, 15] and thus were ineffective against adaptive attacks. As a consequence, adversarial training [44] and its variants [1, 6, 32, 52, 37, 58, 63, 64] have in practice become the mainstream approach to obtain robust models. Specifically, given a dataset $\{(\mathbf{x}_i, y_i)\}_{i=1}^N$, a model $f$ parameterized by $\mathbf{w}$ and a loss function $\mathcal{L}$, adversarial training solves the min-max optimization problem:

$$\min_{\mathbf{w}} \frac{1}{N} \sum_{i=1}^N \max_{\Delta_i \in \mathcal{S}_\epsilon} \mathcal{L}(f(\mathbf{w}, \mathbf{x}_i + \Delta_i), y_i) . \tag{1}$$

In practice, this is achieved by first generating adversarial examples $\mathbf{x}_i + \Delta_i$, usually by PGD, and then using these examples to train the model parameters.

While effective, adversarial training was shown to require a larger model capacity [44, 60]. Specifically, as the model capacity decreases, adversarial training first fails to converge while the training on clean inputs still yields non-trivial performance. Conversely, as the model capacity increases, the performance of training on clean inputs saturates before that of adversarial training. This highlights the challenge of finding robust yet compact models. Here, we introduce a solution to this problem.

**Model Compression.** There are many ways to compress deep neural networks to achieve lower memory consumption and faster inference, including pruning, quantization, and parameter encoding. The pioneering works used information-theoretic methods [39] or second-order derivatives [29] to compress models by removing unimportant weights. The seminal work [28] proposed to prune the parameters with the smallest absolute values for deep networks. This motivated many follow-up works, performing either irregular pruning [26, 61, 65], which removes individual parameters, or regular pruning [31, 42], which aims to discard entire convolutional kernels. In contrast to pruning, quantization [67, 48, 34] seeks to reduce the memory consumption and inference time by using low-precision parameters. An extreme case of quantization is binarization, which can take the

form of only binarizing the parameters [10, 11] or binarizing both parameters and intermediate activations [33]. The models can be further compressed by combining pruning with quantization and Huffman coding [27].

Recently, some efforts have been made to incorporate adversarial training into model compression. [23, 40, 50] suggest quantization as a defense against adversarial attacks. [46] uses Bayesian connectivity sampling to prune the network while preserving its robustness. [8] dynamically generates a robust subnetwork during adversarial training. [62] uses the alternating direction method of multipliers (ADMM) to alternatively conduct adversarial training and network pruning. [25] extends this framework to include other model compression techniques, such as quantization. Furthermore, [54] introduces the HYDRA framework, which improves the performance of compressed robust models by a three-phase method: Pretraining, score-based pruning, and fine-tuning. Here, we follow a different strategy: Instead of performing adversarial training, we search for a robust binary subnetwork in a randomly-initialized one. We show that our approach outperforms those based on adversarial training.

**Lottery Ticket Hypothesis.** This hypothesis, introduced in [21], states that overparameterized neural networks contain sparse subnetworks that can be trained in isolation to achieve competitive performance. These subnetworks are called the *winning tickets*. Based on this interesting observation, [68, 51] further proposed the *Strong Lottery Ticket Hypothesis*. They showed that there exist winning tickets with competitive performance even without training. Furthermore, [9] proposed an iterative randomization scheme to reduce the size of the network in which one searches for the winning tickets. [18] introduced the *Multi-Prize Lottery Ticket Hypothesis* to learn compact yet accurate binary networks by pruning and quantizing randomly weighted DNNs. [17] showed that "lottery-ticket style" approaches can also improve robustness against corruption in the frequency domain.

The recent work of [22] combines robustness with the *Strong Lottery Ticket Hypothesis* and demonstrates the existence of robust sub-networks within a random network. Here, we focus on lighter-weight binary networks, and introduce an adaptive pruning strategy and last batch normalization layer to achieve higher pruning rates than [22] while maintaining a competitive accuracy.

## 3 Methodology

### 3.1 Preliminaries: *Edge-Popup* under Adversarial Attacks

Let us first formulate the problem in a similar manner to [22]. We consider a neural network $f$ parameterized by $\mathbf{w} \in \mathbb{R}^n$. For an input sample $(\mathbf{x}, y)$, the neural network outputs $f(\mathbf{w}, \mathbf{x})$. $\mathcal{L}(f(\mathbf{w}, \mathbf{x}), y)$ then represents the training loss objective, where $\mathcal{L}$ is the softmax cross-entropy loss. Given a dataset $\{(\mathbf{x}_i, y_i)\}_{i=1}^N$, an adversarial budget $\mathcal{S}_\epsilon$, and a predefined pruning rate $r$, we search for a binary pruning mask $\mathbf{m}$ that solves the following optimization problem:

$$\min_{\mathbf{m}} \frac{1}{N} \sum_{i=1}^N \max_{\Delta_i \in \mathcal{S}_\epsilon} \mathcal{L}\left(f(\mathbf{w} \odot \mathbf{m}, \mathbf{x}_i + \Delta_i), y_i\right) \ s.t. \ \mathbf{m} \in \{0, 1\}^n, sum(\mathbf{m}) = (1 - r)n. \quad (2)$$

Here, function $sum$ calculates the summation of all the elements in a vector. In contrast to adversarial training, we do not optimize the model parameters $\mathbf{w}$ in (2); instead $\mathbf{w}$ contains randomly-initialized parameters that are kept fixed during optimization. As such, our algorithm aims to find a pruned network structure, encoded via the $n$-dimensional binary vector $\mathbf{m}$, corresponding to a robust subnetwork. Since the mask $\mathbf{m}$ is a discrete vector, it cannot be directly optimized by gradient-based methods. To overcome this, we replace it with a continuous "score" variable, $\mathbf{s} \in \mathbb{R}^n$, from which we calculate the mask as

$$\mathbf{m} = M(\mathbf{s}, r) , \quad (3)$$

where $M$ is a binarization function. It constructs a binary mask from the continuous-valued scores based on a pruning strategy and a required pruning rate $r$. The pruning strategy can be global or layer-wise, and retains the parameters with the highest scores. In the layer-wise case, it automatically determines the number of parameters retained in each layer.

To update the scores $\mathbf{s}$, we use the same *edge-popup* strategy as in [22, 49]. Specifically, we use straight through estimation [4] to calculate the gradient $\partial\mathcal{L}/\partial\mathbf{s}$. Note that the approximation made in straight through estimation does not affect the adversarial example generation in adversarial training. Our experiments will show that we can effectively generate adversarial examples by PGD. We provide the pseudo-code of our algorithm in Appendix C.

## 3.2 Adaptive Pruning

As mentioned above, in addition to the given pruning rate $r$, the binarization function $M$ in Equation (3) also depends on the pruning strategy. [25] uses *global pruning*, retaining the $(1-r)n$ parameters with the highest scores, regardless of which layer they belong to. However, global pruning does not consider the topology of the network, and the fact that the magnitude of the scores $\mathbf{s}$ can differ from layer to layer. Furthermore, when the pruning rate $r$ is close to 1, global pruning may prune some layers entirely, thus causing a trivial performance. Therefore, other works [22, 51, 54] use layer-wise pruning strategies. For an $L$-layer network with $\{n_i\}_{i=1}^{L}$ parameters and a predefined pruning rate $r$, such strategies first allocate the number of parameters $\{m_i\}_{i=1}^{L}$ to retain in each layer, and then retain the parameters with the highest scores in each layer.

In Appendix A.1, we discuss two special cases of layer-wise pruning: *fixed pruning rate* and *fixed number of parameters*. With *fixed pruning rate*, we have $1-r = \frac{m_1}{n_1} = \frac{m_2}{n_2} = ... = \frac{m_L}{n_L}$. Theorem A.1 indicates that this maximizes the size of the search space of the subnetwork. However, this strategy might retain too few parameters for the small layers when $r$ is big, which has two serious drawbacks: 1) It greatly limits the expression power of the network; 2) it makes the *edge-popup* algorithm less stable, because adding or removing a single parameter then has a large impact on the network's output. This instability becomes even more pronounced in the presence of adversarial samples, because the gradients of the model parameters are more scattered than when training on clean inputs [41].

With *fixed number of parameters*, we have $m_1 = m_2 = ... = m_L$. When the allocated number of retained parameters exceeds the total number of the original parameters in one layer, we leave this layer totally unpruned. As shown in Theorem A.2, this strategy maximizes the number of paths from the input layer to the output layer. In contrast to the *fixed pruning rate*, a *fixed number of parameters* may retain too many parameters in small layers. In the extreme case, some layers may be entirely unpruned when the pruning rate $r$ is small. This is problematic in our settings, since the model parameters are random and not updated.

In other words, the two strategies discussed above are two extremes: the *fixed pruning rate* one suffers when $r$ is big, whereas the *fixed number of parameters* one suffers when $r$ is small. To address this, we propose a strategy in-between these two extremes. Specifically, we determine the number of parameters retained in each layer by solving the following system of equations:

$$1 - r = \frac{\sum_{i=1}^{L} m_i}{\sum_{i=1}^{L} n_i}, \frac{m_1}{n_1^p} = \frac{m_2}{n_2^p} = ... = \frac{m_L}{n_L^p} , \tag{4}$$

where $p \in [0, 1]$ is a hyper-parameter controlling the trade-off between the two extreme cases. When $p = 0$, the strategy (4) is the *fixed number of parameters* one. When $p = 1$, the strategy becomes the *fixed pruning rate* one. By setting $0 < p < 1$, we can retain a higher proportion of parameters in the smaller layers without sacrificing the big layers too much. We call this strategy *adaptive pruning*. As discussed above, the strategy obtained with $p = 1$ tends to fail with a big $r$, while the strategy resulting from setting $p = 0$ tends to fail with a small $r$. This indicates that we need to assign small values of $p$ given a big $r$ and big values of $p$ otherwise. We validate this and study the influence of $p$ on the results of our approach in experiments.

## 3.3 Binary Initialization and Last Normalization Layer

Our work focuses on binary networks, which have a much smaller memory footprint than full-precision networks but are more challenging to train. To address this, we therefore study the influence of the binary initialization scheme and introduce a *last normalization layer* approach to facilitate training and boost the performance.

**Binary initialization.** The empirical studies of [51] demonstrate the importance of the initialization scheme on the performance of a pruned network. As a result, [51] proposes the *Signed Kaiming Constant* initialization: The parameters in layer $i$ are uniformly sampled from the set $\left\{ -\sqrt{\frac{2}{l_{i-1}(1-r)}}, \sqrt{\frac{2}{l_{i-1}(1-r)}} \right\}$, where $l_{i-1}$ represents the fan-out of the previous layer. Correspondingly, the scores $\mathbf{s}$ are initialized based on a uniform distribution $U[-\sqrt{\frac{1}{l_{i-1}}}, \sqrt{\frac{1}{l_{i-1}}}]$.

The magnitude of the *Signed Kaiming Constant* initialization is carefully calculated to keep the variance of the intermediate activations stable from the input to the output. In modern deep neural

networks, the convolutional layers, potentially together with activation functions, are typically followed by a batch normalization layer. In [51] and our settings, these batch normalization layers only estimate the running statistics of their inputs, they do not have trainable parameters representing affine transformations. Because of these batch normalization layers, the magnitudes of the convolutional layers do not affect the outputs of the "convolution-batch norm" blocks. Furthermore, the fully-connected layers on top of the convolutional ones are homogeneous[2] because their bias terms are always initialized to zero and not updated during training. The activation functions we use, such as ReLU or leaky ReLU [43], are also homogeneous. Therefore, the magnitudes of parameters in these fully-connected layers do not change the predicted labels of the model either.

Based on the analysis above, we conclude that the magnitudes of the model parameters at initialization do not change the predicted labels. Therefore, we propose to scale the model parameters $\mathbf{w}$ in all linear layers, i.e., convolutional and fully-connected ones, so that they are all sampled from $\{-1, +1\}$. Correspondingly, the scores $\mathbf{s}$ are initialized based on a uniform distribution $[-a, a]$, where $a$ is a factor controlling the variance.

Our binary initialization scheme is beneficial to model compression and acceleration, since there are no longer multiplication operations in linear layers. We discuss the efficiency improvement of the binary networks in detail in Appendix A.2. Theoretically, for the RN34 models we use in this paper, binary initialization can save approximately $45\%$ and $32\%$ FLOP operations compared with their full precision counterparts in the training phase and evaluation phase, respectively. Since we use irregular pruning in our method, taking full advantage of this improvement requires lower-level and hardware customization.

**Last normalization layer.** Although scaling the model parameters does not affect the expression power of the network nor change the predicted label given the input, it does change the optimization landscape of the problem (2), because the softmax cross-entropy function used to calculate the loss objective is not homogeneous. Compared with the *Signed Kaiming Constant* method, our binary initialization multiplies the parameters initialized in the last layer by $\sqrt{\frac{l_{L-1}(1-r)}{2}}$. Therefore, the output logits fed to the softmax cross-entropy function are also multiplied by the same factor. In practice, $\sqrt{\frac{l_{L-1}(1-r)}{2}} \gg 1$ greatly increases the output logits. Large logits will cause numerical instability and thus greatly worsen the optimization performance. In particular, our detailed analysis in Appendix A.3 shows that such scaling causes gradient vanishing for correctly classified inputs and, even worse, gradient exploding for misclassified ones.

To address this issue, we add another 1-dimensional batch normalization layer at the end of the model, just before the softmax layer. The analysis in Appendix A.3 shows this normalization layer cancels out the multiplication factor applied to the weights in the last layer and thus facilitates the optimization. In our experiments, we show that this normalization layer greatly improves the performance of both the *Signed Kaiming Constant* method and our *Binary Initialization* one. Furthermore, the last normalization layer also makes the performance more robust to different score $\mathbf{s}$ initializations.

## 4 Experiments

In this section, we present extensive experimental results to validate our approach. First, we describe an ablation study and sensitivity analysis. Then, we compare our performance with existing works, which achieve robustness and compression in either full-precision or binary cases. We also include adversarial training [44] as a baseline. Finally, we analyze the structure of the pruned networks that we obtain. We show some interesting patterns of these post-pruning networks, suggesting the potential of our approach for more effective compression.

Unless explicitly stated otherwise, we use a 34-layer Residual Network (RN34) [30], the same as the one in [51, 54].[3] We use the CIFAR10 dataset [36] in the ablation study; we also use the CIFAR100 dataset [36] and the ImageNet100 dataset [16, 20] in the comparisons with the baselines. [4] We train

---

[2]We call a function $f$ homogeneous if it satisfies $\forall x \ \forall a \in \mathbb{R}^+, \ f(ax) = af(x)$.

[3]Note that the RN34 used in these papers and ours differs from the WideRN34-10 used in [44, 58], which is larger and has almost twice the number of trainable parameters.

[4]All these datasets are free for non-commerical use. CIFAR10 and CIFAR100 are downloadable on PyTorch. ImageNet can be downloaded from Kaggle and the subset we use can be found in Contunuum's documentation.

the models for $400$ epochs on CIFAR10/100 and $100$ epochs on ImageNet100. We use a cosine annealing learning rate scheduler with an initial value of $0.1$. Unless specified, we employ PGD attacks [44] to generate adversarial examples during training, but we use AutoAttack (AA) [15] for our robustness evaluation. While PGD is much faster than AutoAttack and thus suitable for training, AutoAttack is the current state-of-the-art attack method, and we thus consider it a more reliable metric of robustness. We use an $l_\infty$ norm-based adversarial budget, and the perturbation strength $\epsilon$ is $8/255$ for CIFAR10, $4/255$ for CIFAR100 and $2/255$ for ImageNet100. More details about the experimental settings and hyper-parameters are listed in Appendix D.1.

## 4.1 Ablation Study and Sensitivity Analysis

**Pruning Strategy and Pruning Rates.** We first focus on binary initialization and on the models with the last batch normalization layer (LBN). We compare the performance of our method under different pruning rates $r$ and *adaptive pruning strategies* with different values of $p$. The scores $\mathbf{s}$ are initialized from a uniform distribution $U[-0.01, 0.01]$.

Our results are summarized in Table 1, in which we include 7 different values of pruning rate $r$ and 7 different values of $p$ in the *adaptive pruning strategy*. First, we notice that the best performance is achieved when $r = 0.99$ and $p = 0.1$. For the *fixed pruning rate* strategy ($p = 1.0$), the best performance is achieved when $r = 0.8$. Compared with the vanilla (i.e., non-adversarial) case in [51], which uses the *fixed pruning rate* strategy and shows that $r = 0.5$ achieves the best clean accuracy, the best performance for robust accuracy is achieved at a much higher pruning rate. This interesting observation is also consistent with the existing work [12], which shows that adversarial training implicitly encourages sparse convolutional kernels.

| Prune Strategy | $r = 0.5$ | $r = 0.8$ | $r = 0.9$ | $r = 0.95$ | $r = 0.99$ | $r = 0.995$ | $r = 0.998$ |
|---|---|---|---|---|---|---|---|
| $p = 0.0$ | 2.16 | 6.86 | 23.01 | 41.61 | 44.60 | 40.70 | **34.97** |
| $p = 0.1$ | 4.35 | 15.03 | 28.12 | 42.65 | **44.88** | **40.97** | 33.09 |
| $p = 0.2$ | 8.01 | 19.21 | 27.99 | 43.72 | 42.92 | 40.52 | 32.99 |
| $p = 0.5$ | 9.21 | 32.70 | 42.84 | 43.62 | 42.45 | 40.55 | 30.08 |
| $p = 0.8$ | 28.90 | 41.51 | **43.64** | **43.88** | 39.12 | 33.61 | 28.07 |
| $p = 0.9$ | 39.09 | 41.71 | 43.07 | 42.28 | 38.68 | 33.89 | 17.43 |
| $p = 1.0$ | **42.85** | **43.23** | 42.13 | 41.12 | 34.57 | 26.67 | 20.56 |

Table 1: Robust accuracy (in %) on the CIFAR10 test set under different pruning rates $r$ and values of $p$ in *adaptive pruning*. The best result for each pruning rate is marked in bold.

Table 1 further demonstrates the benefits of our *adaptive pruning strategy*. For larger pruning rates $r$, a smaller value of $p$ prevails; for smaller pruning rates, a bigger value of $p$ prevails. This is consistent with our analysis in Section 3.2. In particular, compared with the best results for a fixed pruning rate strategy ($p = 1.0$, $r = 0.8$), which is the pruning strategy used in [51], our best adaptive pruning ($p = 0.1$, $r = 0.99$) achieves not only better performance but also a higher pruning rate. That is to say, using our *adaptive pruning strategy* improves both robustness and compression rates.

In Figure 4 of Appendix D.2.6, we provide the learning curves when $r = 0.99$ and when $r = 0.5$. Regardless of the pruning rate $r$, these curves indicate the importance of the pruning strategy: a well chosen $p$ value not only improves the performance but also makes training more stable.

**Last Normalization Layer.** We then study how the last batch normalization layer (LBN) introduced in Section 3.3 affects the performance. We focus on the *binary initialization* first and report the performance of models with and without the last normalization layer under different values of $a$, the hyper-parameter controlling the variance of the initial score $\mathbf{s}$. Based on the results in Table 1, we use the *adaptive pruning strategy* with $p = 0.1$ and a pruning rate $r = 0.99$.

The results are provided in Table 2 and clearly show that the last batch normalization layer (LBN) greatly improves the performance. Furthermore, LBN makes the performance much less sensitive to the initialization of the scores, which in practice facilities the hyper-parameter selection.

**Initialization Scheme.** Finally, we compare the performance of the *binary initialization* with the *Signed Kaiming Constant*. We fix the pruning rate to $r = 0.99$ and employ an adaptive pruning strategy with different values of $p$. Our results are summarized in Table 3. For binary initialization,

| Value of $a$ in in Score Initialization | no LBN | LBN |
|---|---|---|
| 0.001 | 33.08 | **45.06** |
| 0.01 | 39.96 | 44.88 |
| 0.1 | **41.01** | 44.63 |
| 1 | 31.04 | 44.41 |

Table 2: Robust accuracy (in %) on the CIFAR10 test set for models with and without the last batch normalization layer (LBN) under different values of $a$ for score **s** initialization. The best results are marked in bold.

| Prune Strategy | Signed KC | | Binary | |
|---|---|---|---|---|
| | no LBN | LBN | no LBN | LBN |
| $p = 0.0$ | 39.38 | 42.83 | 40.94 | 44.65 |
| $p = 0.1$ | 39.62 | 45.01 | **41.01** | **45.06** |
| $p = 0.2$ | 36.66 | **45.04** | 37.85 | 41.58 |
| $p = 0.5$ | **39.98** | 42.64 | 40.61 | 39.95 |
| $p = 0.8$ | 37.96 | 41.71 | 35.15 | 38.95 |
| $p = 0.9$ | 34.75 | 40.14 | 35.64 | 35.81 |
| $p = 1.0$ | 36.88 | 39.32 | 30.02 | 30.62 |

Table 3: Robust accuracy (in %) on the CIFAR10 test set with the *Signed Kaiming Constant* (Signed KC) and the binary initialization. We include models both with and without the last batch normalization layer (LBN). The best results are marked in bold.

we use the optimal initialization scheme of the score **s** from Table 2; for *Signed Kaiming Constant* initialization, we use the optimal setting from [51] to initialize **s**.

Based on the results in Table 3, we can conclude that the *binary initialization* achieves a comparable performance with the *Signed Kaiming Constant*. Furthermore, the last batch normalization layer also improves the performance when using the *Signed Kaiming Constant*. We show in Table 8 of Appendix D.2.1 that these conclusions are also valid in a non-adversarial setting.

## 4.2 Comparison with Existing Methods

**Baselines.** In this section, we compare our approach with the state-of-the-art methods targeting model compression and robustness. Specifically, we include FlyingBird, FlyingBird+[8], Bayesian Connectivity Sampling (BCS) [46], Robust Scratch Ticket (RST) [22], HYDRA [54] and ATMC [25], as well as adversarial training (AT) [44] with early stopping [53]. Given our previous results, we fix the pruning rate to $r = 0.99$. For adversarial training, we use the full RN34 model and some smaller networks with approximately the same number of parameters as our pruned models. These smaller networks have the same architecture as the RN34 except that they have fewer channels. The details of these small networks are shown in Table 7 of Appendix D.1. We follow the official implementations of all the baselines, and thus, unlike in our method, the normalization layers in all the baselines that update model parameters have an affine transformation with trainable parameters.

ATMC supports quantization but its parameterization introduces learnable quantized values. That is, although the models obtained by ATMC's 1-bit quantization have only two parameter values in each layer, these values are different from layer to layer and are not necessarily $-1$ and $+1$. This means that, compared with the binary networks obtained with our method, those from ATMC have more trainable parameters and thus flexibility. Nevertheless, we still include ATMC for comparison in the case of binary networks. Similarly to our method, RST does not update the model parameters. It initializes the model parameters with full-precision values, and we thus only provide full-precision results for RST. The other baselines and AT are not designed for quantization and do not inherently support binary networks. To address this, we use *BinaryConnect* [10] to replace the model's linear layers so that their parameters are binary. *BinaryConnect* generates binarized model parameters by taking the sign of the weights during the forward pass, and uses straight-through estimation [4] for gradient calculation.

Our method uses *binary initialization* and the last batch normalization layer, so the models we obtained are inherently binary. In addition to using PGD-based adversarial examples, we accelerate our method by using adversarial examples based on FGSM [24] with ATTA [66]. FGSM with ATTA generates adversarial examples by one-step attacks with accumulated perturbations across epochs. This is much cheaper than the 10-step PGD attacks. For CIFAR10 and CIFAR100, we provide the results of our method when using FGSM with ATTA as "Ours(fast)" in Table 4 for comparison. For ImageNet100, since the dataset is bigger and the images are of much higher resolution, the computational cost for multi-step PGD is huge. Therefore, we use FGSM with ATTA to generate adversarial examples for all methods on ImageNet100. To decrease the memory overhead introduced

| Method | Architecture | Pruning Strategy | CIFAR10 FP | CIFAR10 Binary | CIFAR100 FP | CIFAR100 Binary | ImageNet100 FP | ImageNet100 Binary |
|---|---|---|---|---|---|---|---|---|
| AT | RN34 | Not Pruned | 43.26 | 40.34 | 36.63 | 26.49 | 53.92 | 34.20 |
| AT | RN34-LBN | Not Pruned | 42.39 | 39.58 | 35.15 | 32.98 | 55.14 | 35.36 |
| AT | Small RN34 | Not Pruned | 38.81 | 26.03 | 27.68 | 15.85 | 25.40 | 10.44 |
| FlyingBird | RN34 | Dynamic | 45.86 | 34.37 | 35.91 | 23.32 | 37.70 | 9.54 |
| FlyingBird+ | RN34 | Dynamic | 44.57 | 33.33 | 34.30 | 22.64 | 37.70 | 9.52 |
| BCS | RN34 | Dynamic | 43.51 | - | 31.85 | - | - | - |
| RST | RN34 | $p = 1.0$ | 34.95 | - | 21.96 | - | 17.54 | - |
| RST | RN34-LBN | $p = 1.0$ | 37.23 | - | 23.14 | - | 15.36 | - |
| HYDRA | RN34 | $p = 0.1$ | 42.73 | 29.28 | 33.00 | 23.60 | 43.18 | 18.22 |
| ATMC | RN34 | Global | 34.14 | 25.62 | 25.10 | 11.09 | 22.18 | 5.78 |
| ATMC | RN34 | $p = 0.1$ | 34.58 | 24.62 | 25.37 | 11.04 | 23.52 | 4.58 |
| Ours | RN34-LBN | $p = 0.1$ | - | **45.06** | - | **34.83** | - | **33.04** |
| Ours(fast) | RN34-LBN | $p = 0.1$ | - | 40.77 | - | 34.45 | | |

Table 4: Robust accuracy (in %) on the CIFAR10, CIFAR100 and ImageNet100 test sets for the baselines and our proposed method. "RN34-LBN" represents ResNet34 with the last batch normalization layer. "Small RN34" refers to Small RN34-p0.1 in Table 7 of Appendix D.1. The pruning rate is set to 0.99 except for the not-pruned methods. Among the pruned models, the best results for the full-precision (FP) models are underlined; the best results for the binary models are marked in bold. The values of $\epsilon$ for CIFAR10, CIFAR100 and ImageNet100 are $8/255$, $4/255$ and $2/255$, respectively. "-" means not applicable or trivial performance.

by ATTA, we only store the downsampled perturbations in the current epoch for the perturbation initialization of the next epoch. We provide the pseudo-code and more details in Appendix C.

**Results.** Our main results on CIFAR10, CIFAR100 and ImageNet100 are summarized in Table 4, where we report the robust accuracy under AutoAttack (AA), which is considered as a reliable evaluation metric for robustness [15]. The results of all baselines are based on their default settings in architecture and pruning strategy based on publicly available codes. [5] The exceptions are that we also include adaptive pruning ($p = 0.1$) for HYDRA, ATMC, and the last batch normalization layer for RST, because we noticed such changes to improve their performance.

Our method using the *adaptive pruning* strategy ($p = 0.1$) achieves better performance than all baselines in case of binary models. On CIFAR10 and CIFAR100, we also achieve comparable performance to methods using full-precision models. Furthermore, our method achieves results comparable with AT on the original unpruned models that has $100\times$ more trainable parameters. In addition, our method based on FGSM with ATTA, which is much faster than multi-step PGD, also achieves better performance than all baselines in the case of binary networks. On ImageNet100, our method, which aims to train binary networks, also outperforms most full-precision networks trained by the baselines. BCS yields almost trivial performance on ImageNet100 ($< 3\%$) and is thus not included. This suggests that BCS cannot converge using a high compression rate and facing a complicated dataset. Compared with adversarial training on the full network, which has 100 times as many parameters as ours, we achieve comparable performance with the binary networks, but worse performance than the full-precision networks. Note that fitting the high-dimensional ImageNet100 dataset under adversarial attacks using only 1% of the binary parameters is extremely challenging. As demonstrated in Table 4, many baselines only achieve low robust accuracy in this setting.

For all baselines except RST, the last normalization layer does not improve the performance; it even hurts the performance in the full-precision cases. This is because these baselines (except RST) update the model parameters $\mathbf{w}$. In the full precision cases, the magnitude of $\mathbf{w}$, and thus of the output logits, is automatically adjusted during training. The issue resulting from large output logits that we pointed out in Section 3 does thus not happen in these cases, so the last batch normalization layer is not necessary. In practice, we observed this layer to slow down the training convergence of these models.

---

[5]Publicly available code on GitHub: FlyingBird/FlyingBird+: VITA-Group/Sparsity-Win-Robust-Generalization; BCS: IGITUGraz/SparseAdversarialTraining; RST: RICE-EIC/Robust-Scratch-Ticket; HYDRA: inspire-group/hydra; ATMC: VITA-Group/ATMC. All the codes are free to use for non-commerical purposes.

For the pruning strategy, the proposed *adaptive pruning* strategy ($p = 0.1$) consistently achieves better performance than the *fix pruning rate* strategy ($p = 1.0$) and than *global pruning*. FlyingBird, FlyingBird+ and BCS dynamically assign retrained parameters during training, which has similar benefits to adaptive pruning but at the cost of training efficiency [8]. Furthermore, although the value of $p$ is selected based on the ablation study on CIFAR10, it also performs well on CIFAR100 and ImageNet100. This observation indicates that for a fixed value of $r$, the selection of $p$ generalizes well across different datasets.

We further compare the baseline methods with various settings such as adding the last batch normalization layer, changing the pruning strategy, using different AT methods, and provide a complete set of comparison results in Appendix D.2.2. The conclusions drawn from Table 4 remain valid.

| Method | Architecture | Pruning Strategy | RN18 FP | RN18 Binary | RN50 FP | RN50 Binary |
|--------|--------------|------------------|---------|-------------|---------|-------------|
| AT | RN | Not Pruned | 41.50 | 39.13 | 43.24 | 31.18 |
| AT | RN-LBN | Not Pruned | 42.25 | 39.86 | 44.33 | 37.25 |
| AT | Small RN | Not Pruned | 28.13 | 30.35 | 26.03 | 32.25 |
| FlyingBird | RN | Dynamic | 42.15 | 27.08 | 35.91 | 26.33 |
| FlyingBird+ | RN | Dynamic | 38.55 | 27.84 | 29.54 | 25.40 |
| BCS | RN | Dynamic | 39.60 | 21.46 | 41.85 | 17.54 |
| RST | RN | $p = 1.0$ | 31.98 | - | 35.40 | - |
| RST | RN-LBN | $p = 1.0$ | 33.27 | - | 34.71 | - |
| HYDRA | RN | $p = 0.1$ | 40.20 | 30.90 | 44.14 | 22.36 |
| ATMC | RN | Global | 32.21 | 17.73 | 25.23 | 6.82 |
| ATMC | RN | $p = 0.1$ | 32.31 | 19.67 | 33.61 | 16.12 |
| Ours | RN-LBN | $p = 0.1$ | - | **39.65** | - | **42.72** |
| Ours (fast) | RN-LBN | $p = 0.1$ | - | 30.86 | - | 37.93 |

Table 5: Robust accuracy (in %) on the CIFAR10 test set for AT, FlyingBird(+), BCS, RST, HYDRA, ATMC and our proposed method on the RN18 and RN50 models. "RN-LBN" represents networks with the last batch normalization layer. Among the compressed models, the best results for full precision (FP) models are underlined; the best results for binary models are marked in bold.

All the results in Table 4 are based on a RN34 architecture, Table 5 provides the results on CIFAR10 using a smaller 18-layer network (RN18) and a larger 50-layer network (RN50). These results confirm the effectiveness of our method on different network architectures.

In addition to robust accuracy, Table 10 in Appendix D.2.3 demonstrates the accuracy on the clean test set for the models in Table 4. Our method also yields competitive performance on clean inputs. Specifically, we achieve the best performance among all methods for binary networks. Combining the results in Table 4 and 10, we conclude that our method yields a better trade-off between accuracy on clean inputs and accuracy on adversarially perturbed inputs.

Finally, vanilla training can be considered as a special case of adversarial training, where $\epsilon = 0$. Therefore, our method, as well as all baselines, are applicable to vanilla training. The results when $\epsilon = 0$ are provided in Table 11 of Appendix D.2.4. Our method achieves the best performance among the pruned binary networks. This indicates that our method is competitive under difference adversarial budgets.

### 4.3 Analysis of the Subnetwork Patterns

In this work, we use irregular pruning. Compared with regular pruning, irregular pruning is more flexible but less structured, which means that it requires lower-level customization to fully take advantage of parameter sparsity for acceleration. However, visualizing the masks $\mathbf{m}$ of the convolutional layers in our pruned binary network with a pruning rate $r = 0.99$ allowed us to find that the mask is structured to some degree. For example, we visualize the mask of a convolutional layer with 256 input channels and 256 output channels in Figure 5 of Appendix D.2.5. We notice that the retained parameters are quite concentrated and structured: Most retained parameters concentrate on few input or output channels, while many other channels ($40\%$ of the total) are completely pruned.

Furthermore, we visualize two consecutive convolutional layers in the same residual block of the RN34 model. We call them *layer1* and *layer2* following the forward pass. In Figure 1, we plot the distribution of the retained parameters in each input channel and in each output channel, respectively. We find that many output channels of layer1 and input channels of layer2, 40% of all channels in this case, are totally pruned. As a reference, we also plot the distribution of random pruning, based on the average of 500 simulations. As demonstrated in Figure 1, the distribution of the retained parameters in each channel is much more uniform in this case. Our theoretical analysis in Appendix A.4 demonstrates that, in a randomly pruned network, it is almost impossible to have even one entirely pruned channel. The comparison indicates that the mask **m** obtained by our method is structured.

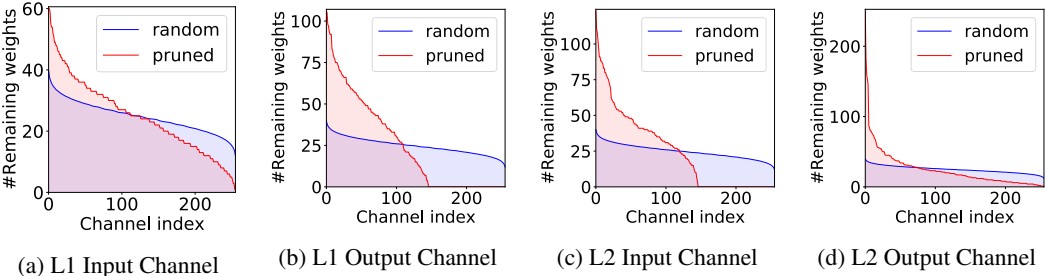

| (a) L1 Input Channel | (b) L1 Output Channel | (c) L2 Input Channel | (d) L2 Output Channel |

Figure 1: Number of retained parameters in each input and output channel of layer1 (L1) and layer2 (L2) in the same residual block. We sort the numbers and plot the curves from the largest on the left to the smallest on the right. The red curves represent the mask obtained by our method; the blue curves depict what happens when randomly pruning the corresponding layer.

The observations in Figure 1 also hold for kernels: a few kernels, of size $3 \times 3$ and thus having 9 entries, are totally unpruned. We show the distribution of the number of retained parameters in each kernel in Figure 2 and provide the distribution by random uniform pruning as a reference. Random uniform pruning yields no kernels with more than 3 retained parameters, but many such kernels can be observed in the masks generated by our method. Finally, in Figure 3 of Appendix D.2.5, we visualize the positions of the pruned output channels of layer1 and the pruned input channels of layer2. We observe those pruned channels to be aligned. That is, some neurons representing both the output channels of layer1 and the input channels of layer2 are entirely removed. We defer additional discussions, figures and results to Appendix D.2.5. The pattern of the structures learned by our method indicates the potential of regular pruning for a randomly-initialized network in the presence of adversarial attacks. We leave this as future work.

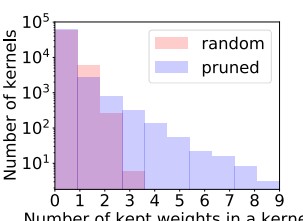

Figure 2: Distribution of the number of retained parameters in each kernel. The y-axis is in log-scale.

## 5   Conclusion

We have proposed a method to obtain robust binary models by pruning randomly-initialized networks, thus extending the *Strong Lottery Ticket Hypothesis* to the case of robust binary networks. In contrast to the state-of-the-art methods, we learn the structure of robust subnetworks without updating the parameters. Furthermore, we have proposed an *adaptive pruning* strategy and last batch normalization layer to stabilize the training and improve performance. Finally, we have relied on binary initialization to obtain more compact models.

Our extensive results on various benchmarks have demonstrated that our approach outperforms existing methods for training compressed robust models. Furthermore, we have observed interesting structured patterns occurring in the parameters retained in the subnetworks. This opens the door to further investigations on the structure of the robust subnetworks and on the design of regular pruning strategies in the adversarial scenario.

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
