# A Analysis

## A.1 Analysis of Layerwise Pruning Strategies

### A.1.1 Fixed Pruning Rate

We consider an $L$-layer neural network and each layer has $n_1$, $n_2$, ..., $n_L$ parameters, we retrain $m_1$, $m_2$, ..., $m_L$ parameters after pruning. As such, the total number of combinations $\Pi_{i=1}^{L} \binom{n_i}{m_i}$ is the size of search space of the subnetworks. The following theorem shows, the *fixed pruning rate* strategy is the strategy which approximates the maximization of the total number of combinations.

**Theorem A.1.** *Consider an $L$-layer neural network with $n_1$, $n_2$, ..., $n_L$ parameters in each layer, we retrain $m_1$, $m_2$, ..., $m_L$ parameters after pruning. Given a predefined pruning rate $r = 1 - \frac{\sum_{i=0}^{L} m_i}{\sum_{i=0}^{L} n_i}$, the optimal numbers of post-pruning parameters $\{m_i\}_{i=1}^{L}$ that maximizing the total number of combinations $\Pi_{i=1}^{L} \binom{n_i}{m_i}$ satisfy the following inequality:*

$$\forall 1 \le j, k \le L, \quad \left| \frac{m_j}{n_j} - \frac{m_k}{n_k} \right| < \frac{1}{n_j} + \frac{1}{n_k} \tag{5}$$

We defer the proof to Appendix B.1. Specifically, we let $n_k$ in (5) be the largest layer in the network without the loss of generality, we then have $\forall i \le j \le L, j \ne k, \left| m_j - \frac{m_k}{n_k} n_j \right| < \frac{n_j}{n_k} + 1 \le 2$. $m_j$ is the number of retained parameters and thus an integer, so Theorem A.1 indicates the pruning rate of each layer is close to each other when we aim to maximize the total number of combinations. Therefore, we can consider the *fixed pruning rate* strategy, i.e., $1 - r = \frac{m_1}{n_1} = \frac{m_2}{n_2} = ... = \frac{m_L}{n_L}$, as an approximation to maximize the total number of combinations.

**Drawbacks** While this strategy may seem intuitive, it does not take the differences in layer size into account. In practice, the number of parameters in different layers can vary widely. For example, residual networks [30] have much fewer parameters in the first and last layers than in the middle ones. Using *fixed pruning rate* thus yields very few parameters after pruning within such small layers. For example, when $r = 0.99$, only 17 parameters are left after pruning for a convolutional layer with 3 input channels, 64 output channels and a kernel size of 3. Such a small number of parameters has two serious drawbacks: 1) It greatly limits the expression power of the network; 2) it makes the edge-popup algorithm less stable, because adding or removing a single parameter then has a large impact on the network's output. This instability becomes even more pronounced in the presence of adversarial samples, because the gradients of the model parameters are more scattered than when training on clean inputs [41].

### A.1.2 Fixed Number of Parameters

To overcome drawbacks of the fixed pruning rate strategy, we study an alternative strategy aiming to maximize the total number of paths from the input to the output in the pruned network. For a feedforward network, the total number of such paths is upper bounded by $\Pi_{i=1}^{L} m_i$. The following theorem demonstrates that the pruning strategy that maximizes this upper bound consists of retaining the same number of parameters in every layer, except for the layers that initially have too few parameters, for which all parameters should then be retained. This optimal strategy is the *fixed number of parameters* mentioned in Section 3.2.

**Theorem A.2.** *Consider an $L$-layer feedforward neural network with $n_1, n_2, ..., n_L$ parameters in its successive layers, from which we retain $m_1, m_2, ..., m_L$ parameters, respectively, after pruning. Given a predefined sparsity ratio $r = 1 - \frac{\sum_{i=1}^{L} m_i}{\sum_{i=1}^{L} n_i}$, the numbers of post-pruning parameters $\{m_i\}_{i=1}^{L}$ that maximize the upper bound of the total number of the input-output paths $\Pi_{i=1}^{L} m_i$ have the following property: $\forall 1 \le j \le L$, $m_j$ satisfies either of the following two conditions: 1) $m_j = n_j$; 2) $\forall 1 \le k \le L, m_j \ge m_k - 1$.*

The two conditions in Theorem A.2 mean we retain the same number of parameters for each layer except for ones totally unpruned. We defer the proof to Appendix B.2, where we use proof by contraction.

**Drawbacks** While this *fixed number of parameters* strategy addresses the problem of obtaining too small layers arising in the *fixed pruning rate* one, it suffers from overly emphasizing the influence of the small layers. That is, the smaller layers end up containing too many parameters. In the extreme case, some layers are totally unpruned when the pruning rate $r$ is small. This is problematic in our settings, since the model parameters are random and not updated. The unpruned layers based on random parameters provide a large amount of noise in the forward process. Furthermore, this strategy significantly sacrifices the expression power of the big layers.

## A.2  Analysis of Acceleration by Binary Initialization

In this section, we analyze the acceleration benefit of binary initialization. Since most of the forward and backward computational complexity for the models studied in this paper is consumed by the "Convolutional-BatchNorm-ReLU" block, the acceleration rate on such blocks is a good approximation of that on the whole network. Therefore, we concentrate on the "Convolution-BatchNorm-ReLU" block here.

For simplicity, we assume the feature maps and convolutional kernels are all squares. Without the loss of generality, we consider $r_{in}$-channel input feature maps of size $s$, the size of the convolutional kernel is $c$ and the convolutional layer outputs $r_{out}$ channels. Here, the binary layers represent the layer whose parameters are either $-1$ or $+1$.

**Forward Pass** For full-precision dense networks, the number of FLOP operations of the convolutional layer is $2c^2s^2r_{in}r_{out}$. By contrast, the complexity can be reduced to $c^2s^2r_{in}r_{out}$ for binary dense layers, convolution operation with a binary kernel does not include any multiplication operations. Correspondingly, for sparse layers whose pruning ratio is $r$, the complexity of full-precision sparse networks and of the binary sparse networks can be reduced to $2(1-r)c^2s^2r_{in}r_{out}$ and $(1-r)c^2s^2r_{in}r_{out}$, respectively.

The batch normalization layer will consume $3s^2r_{out}$ FLOP operations during inference and $10s^2r_{out}$ during training. The additional operations during training are due to the update of running statistics. Note that, the batch normalization layer in our random initialized network does not contain any trainable parameters, so there is no scaling parameters after normalization. The ReLU layer will always consume $s^2r_{out}$ FLOP operations.

To sum up, we calculate the complexity ratio of the binary "Convolution-BatchNorm-ReLU" block over its full-precision counterpart in the forward pass. For dense layers, the ratio is $\frac{c^2r_{in}+11}{2c^2r_{in}+11}$ for the training time and $\frac{c^2r_{in}+4}{2c^2r_{in}+4}$ for the inference. For sparse layers, the ratio is $\frac{(1-r)c^2r_{in}+11}{2(1-r)c^2r_{in}+11}$ for the training time and $\frac{(1-r)c^2r_{in}+4}{2(1-r)c^2r_{in}+4}$ for the inference.

**Backward Pass** Compared with the forward pass, the backward pass has some computational overhead, because we need to calculate the gradient with respect to the score variable **s** associated with the convolutional kernels. For both dense and sparse networks, the overhead is $2c^2s^2r_{in}r_{out} + c^2r_{in}r_{out}$ for full precision layers and $2c^2s^2r_{in}r_{out}$ for binary layers. Note that, the overhead is independent of the pruning rate $r$ because the pruning function is treated as the identity function in the backward pass. In addition, the difference here between the full precision layer and binary layer arises from the multiplication when we backprop the gradient through the weights.

To sum up, we calculate the complexity ratio of the binary "Convolution-BatchNorm-ReLU" block over its full-precision counterpart in the backward pass. We only back propagate the gradient in the training time, so the batch normalization layer is always the training mode. For dense layers, the ratio is $\frac{3c^2s^2r_{in}+4s^2}{4c^2s^2r_{in}+4s^2+c^2r_{in}}$. For sparse layers, the ratio is $\frac{3(1-r)c^2s^2r_{in}+4s^2}{4(1-r)c^2s^2r_{in}+4s^2+c^2r_{in}}$.

| | Full Precision | Binary |
|---|---|---|
| Forward - Training | $2(1-r)c^2s^2r_{in}r_{out} + 11s^2r_{out}$ | $(1-r)c^2s^2r_{in}r_{out} + 11s^2r_{out}$ |
| Forward - Evaluation | $2(1-r)c^2s^2r_{in}r_{out} + 4s^2r_{out}$ | $(1-r)c^2s^2r_{in}r_{out} + 4s^2r_{out}$ |
| Backward - Training | $4(1-r)c^2s^2r_{in}r_{out} + 4s^2r_{out} + c^2r_{in}r_{out}$ | $3(1-r)c^2s^2r_{in}r_{out} + 4s^2r_{out}$ |

Table 6: The complexity in FLOP operations of the sparse "Convolution-BathNorm-ReLU" block in both full precision and binary case. The pruning rate is $r$.

**Discussion** We summarize the complexity in FLOP operations of the sparse "Convolution-BatchNorm-ReLU" block in different scenarios. We can now conclude that compared with the full precision block, the binary block decrease the overall complexity in two places: 1) we save $(1-r)c^2s^2r_{in}r_{out}$ FLOPs for the convolution and transpose convolution operations in the forward and backward pass, respectively; 2) for the backpropagation, we save $c^2r_{in}r_{out}$ FLOPs, because there is no multiplication when we backprop the gradient through the weights for binary blocks.

We consider the practical settings: $r = 0.99$, $c = 3$, $r_{in} = r_{out} = 128$, $s = 16$. The complexity ratio of the binary block over the full precision block in the forward pass is $\frac{(1-r)c^2r_{in}+11}{2(1-r)c^2r_{in}+11} = 0.6616$ for the training mode and $\frac{(1-r)c^2r_{in}+4}{2(1-r)c^2r_{in}+4} = 0.5740$ for the evaluating mode, respectively. The complexity ratio in the backward pass is $\frac{3(1-r)c^2s^2r_{in}+4s^2}{4(1-r)c^2s^2r_{in}+4s^2+c^2r_{in}} = 0.7065$. That is to say, compared with the full precision block, the binary block under this setting can save around $34\%$ and $29\%$ time in the forward and backward passes during training; for inference, it can save $43\%$ time.

## A.3 Analysis of the Normalization Layer before Softmax

We consider a $L$-layer neural network and each layer has $l_1$, $l_2$, ..., $l_L$ neurons. Let $\mathbf{u} \in \mathbb{R}^{l_{L-1}}$, $\mathbf{W} \in \mathbb{R}^{l_L \times l_{L-1}}$, $\mathbf{o} \in \mathbb{R}^{l_L}$ be the output of the penultimate's output, the weight matrix of the last fully-connected layer and the last layer's output, respectively. In addition, we use $c \in \{1, 2, ..., l_L\}$ to denote the label of the data and omit the bias term of the last layer since it is initialized as 0 and is not updated. For the 1-dimensional batch normalization layer, we use $\mathbf{b} \in \mathbb{R}^{l_L}$ and $\mathbf{v} \in \mathbb{R}^{l_L}$ to represent the running mean and running standard deviation, respectively.

Therefore, the loss objective $\mathcal{L}_{wo}$ and its gradient of the model without the 1-dimensional batch normalization layer is:

$$\mathcal{L}_{wo} = -log\frac{e^{\mathbf{o}_c}}{\sum_{i=1}^{l_L} e^{\mathbf{o}_i}}$$
$$\frac{\partial\mathcal{L}_{wo}}{\partial\mathbf{o}_j} = \frac{e^{\mathbf{o}_j}}{\sum_{i=1}^{l_L} e^{\mathbf{o}_i}} - \mathbf{1}(j = c)$$

(6)

Correspondingly, the loss objective $\mathcal{L}_{wi}$ and its gradient of the model with the 1-dimensional batch normalization layer is:

$$\mathcal{L}_{wi} = -log\frac{e^{(\mathbf{o}_c-\mathbf{b}_c)/\mathbf{v}_c}}{\sum_{i=1}^{l_L} e^{(\mathbf{o}_i-\mathbf{b}_i)/\mathbf{v}_i}}$$
$$\frac{\partial\mathcal{L}_{wi}}{\partial\mathbf{o}_j} = \frac{1}{\mathbf{v}_j}\left(\frac{e^{(\mathbf{o}_c-\mathbf{b}_c)/\mathbf{v}_c}}{\sum_{i=1}^{l_L} e^{(\mathbf{o}_i-\mathbf{b}_i)/\mathbf{v}_i}} - \mathbf{1}(j = c)\right)$$

(7)

Now we consider the case when the model parameter $\mathbf{W}$ is multiplied by a factor $\alpha > 1$: $\mathbf{W}' = \alpha\mathbf{W}$ and assume the output of the penultimate layer is unchanged. In practice, $\alpha$ is far more than 1. For example, if the penultimate layer has 512 neurons, $\alpha$ will be 16 when we change kaiming constant initialization to binary initialization. Based on this, the new output of the last layer is $\mathbf{o}' = \alpha\mathbf{o}$. For the model with the normalization layer, the new statistics are $\mathbf{b}' = \alpha\mathbf{b}$ and $\mathbf{v}' = \alpha\mathbf{v}$. In this regard, we can then recalculate the gradient of the loss objective as follows:

$$\frac{\partial\mathcal{L}'_{wo}}{\partial\mathbf{o}'_j} = \frac{e^{\mathbf{o}'_j}}{\sum_{i=1}^{l_L} e^{\mathbf{o}'_i}} - \mathbf{1}(j = c) = \frac{e^{\alpha\mathbf{o}_j}}{\sum_{i=1}^{l_L} e^{\alpha\mathbf{o}_i}} - \mathbf{1}(j = c)$$
$$\frac{\partial\mathcal{L}'_{wi}}{\partial\mathbf{o}'_j} = \frac{1}{\mathbf{v}'_j}\left(\frac{e^{(\mathbf{o}'_c-\mathbf{b}'_c)/\mathbf{v}'_c}}{\sum_{i=1}^{l_L} e^{(\mathbf{o}'_i-\mathbf{b}'_i)/\mathbf{v}'_i}} - \mathbf{1}(j = c)\right) = \frac{1}{\alpha\mathbf{v}_j}\left(\frac{e^{(\mathbf{o}_c-\mathbf{b}_c)/\mathbf{v}_c}}{\sum_{i=1}^{l_L} e^{(\mathbf{o}_i-\mathbf{b}_i)/\mathbf{v}_i}} - \mathbf{1}(j = c)\right)$$

(8)

We first study the case without the normalization layer. The first term $\frac{e^{\alpha\mathbf{o}_j}}{\sum_{i=1}^{l_L} e^{\alpha\mathbf{o}_i}}$ of the gradient $\frac{\partial\mathcal{L}'_{wo}}{\partial\mathbf{o}'_j}$ converge to $\mathbf{1}(j = \text{argmax}_i\mathbf{o}_i)$ exponentially. For correctly classified inputs, $\frac{\partial\mathcal{L}'_{wo}}{\partial\mathbf{o}'_j}$ converge to 0

exponentially with $\alpha$. In addition, the gradient $\frac{\partial \mathcal{L}'_{wo}}{\partial \mathbf{u}} = \mathbf{W}'^T \frac{\partial \mathcal{L}'_{wo}}{\partial \mathbf{o}'_j} = \alpha \mathbf{W}^T \frac{\partial \mathcal{L}'_{wo}}{\partial \mathbf{o}'_j}$ also vanish with $\alpha$.
$\frac{\partial \mathcal{L}'_{wo}}{\partial \mathbf{u}}$ is backward to previous layers, leading to gradient vanishing. For incorrectly classified inputs, $\frac{\partial \mathcal{L}'_{wo}}{\partial \mathbf{o}'_j}$ converge to $\mathbf{1}(j = \mathrm{argmax}_i \mathbf{o}_i) - \mathbf{1}(j = c)$, which is a vector with $c$-th element being $-1$, the element corresponding to the output label being $+1$ and the rest elements being $0$. In this case, the gradient backward $\frac{\partial \mathcal{L}'_{wo}}{\partial \mathbf{u}} = \alpha \mathbf{W}^T \frac{\partial \mathcal{L}'_{wo}}{\partial \mathbf{o}'_j}$ will be approximately multiplied by $\alpha$, causing gradient exploding.

By contrast, in the case of the model with the normalization layer, $\frac{\partial \mathcal{L}'_{wi}}{\partial \mathbf{o}'_j} = \frac{1}{\alpha} \frac{\partial \mathcal{L}_{wi}}{\partial \mathbf{o}_j}$. The factor $\frac{1}{\alpha}$ is cancelled out when we calculate $\frac{\partial \mathcal{L}'_{wi}}{\partial \mathbf{u}} = \mathbf{W}'^T \frac{\partial \mathcal{L}'_{wi}}{\partial \mathbf{o}} = \mathbf{W}^T \frac{\partial \mathcal{L}_{wi}}{\partial \mathbf{o}}$. This means the gradient backward remains unchanged if we use the 1-dimensional batch normalization layer, which maintains the stability of training if we scale the model parameters.

To conclude, the 1-dimensional batch normalization layer is crucial to maintain the stability of training if we use binary initialization. Without this layer, the training will suffer from gradient vanishing for correctly classified inputs and gradient exploding for incorrectly classified inputs.

### A.4 Analysis of the structure of a randomly pruned network

In this section, we provide preliminary analysis of the structure of a randomly pruned network.

As a starting point, we first estimate the probability of $k$ retained parameters in a $3 \times 3$ kernel. Given the pruning rate $r_i$ for the layer $i$ with $n_i$ weights, the number of the retained parameters is $m_i := (1 - r_i)n_i$. We assume $m_i$ lies in a proper range: $9 \ll m_i < \frac{1}{9}n_i$. This is true when $n_i$ is large and $r_i > \frac{8}{9}$.

For each kernel $j$, we use $X_j$ to represent its number of retained parameters. It is difficult to calculate $P(X_j = k)$ directly because $\{X_j\}_j$ are constrained by: 1) $\sum_j X_j = m_i$; 2) $\forall j, 0 \leq X_j \leq 9$. However, in the case of random pruning, we have $E[X_j] = \frac{9m_i}{n_i} = 9(1 - r_i) < 1$. In this regard, we can make the approximation by removing the constraint $X_j \leq 9$.

Therefore, we can reformulate the problem of calculating $P(X_j = k)$ as: *Given $m_i$ steps, randomly select one box out of the total $\frac{n_i}{9}$ boxes and put one apple in it. $P(X_j = k)$ is then the probability for the box $j$ to have $k$ apples.*

In this approximation, it is straightforward to have $P(X_j = k) = \binom{m_i}{k} P_i^k \cdot (1 - P_i)^{m_i - k}, 0 \leq k \leq 9$ where $P_i = \frac{9}{n_i}$. Based on the assumption that $n_i$ is large, $P_i \approx 0$. Therefore, $P(X_j = 0) = (1 - \frac{9}{n_i})^{m_i} \approx e^{9(1 - r_i)}$. For $k > 1$, we apply Stirling approximation $n! \approx \sqrt{2\pi n}(\frac{n}{e})^n$ to the binomial coefficient, then

$$
\begin{aligned}
P(X_j = k) &\approx \frac{m_i^{m_i + 0.5}}{\sqrt{2\pi} k^{k+0.5}(m_i - k)^{m_i - k + 0.5}} \cdot \left(\frac{9}{n_i}\right)^k \cdot \left(1 - \frac{9}{n_i}\right)^{m_i - k} \\
&= \sqrt{\frac{m_i}{2\pi k(m_i - k)}} \cdot \left(\frac{9(1 - r_i)}{k}\right)^k \cdot \left(1 + \frac{k}{m_i - k}\right)^{m_i - k} \cdot \left(1 - \frac{9}{n_i}\right)^{m_i - k}
\end{aligned}
\tag{9}
$$

The second equality is based on the fact $m_i = (1 - r_i)n_i$. Since $m_i \gg 9 > k$ by the assumption and $n_i \gg m_i$, we can approximate $(1 - \frac{9}{n_i})^{m_i - k}$ to $1 - \frac{9(m_i - k)}{n_i} \approx 1$, then

$$
P(X_j = k) \approx \sqrt{\frac{m_i}{2\pi k(m_i - k)}} \cdot \left(\frac{9e(1 - r_i)}{k}\right)^k = \sqrt{\frac{m_i}{2\pi k(m_i - k)}} \cdot \left(\frac{c}{k}\right)^k
\tag{10}
$$

where $c = 9e(1 - r_i)$ is a constant.

As shown in the equation above, $P(X_j = k)$ decreases drastically when $k$ increases. Therefore, in a randomly pruned layer $i$ with $n_i = 3 \times 3 \times 256 \times 256 = 589824$ and $r_i = 0.99$, it is almost impossible to see kernels who have at least 4 retained parameters, because according to the above formula, the estimated number of kernels in that layer having 3 retained parameters is $\frac{n_i}{9} \times P(X_j = 3) \approx 8.19$, and the number of kernels having 4 retained parameters is $\approx 0.18$.

Now we consider the number of retained parameters in a channel. For the layer of $r_{in}$ input channels and $r_{out}$ output channels, it has $r_{in} \times r_{out} \times 3 \times 3$ parameters. We use $Y_j$ to represent the number of the retained parameters for the input channel $j$. Similarly, for the random pruning, we have

$$P(Y_j = k) \approx \sqrt{\frac{m_i}{2\pi k(m_i - k)}} \cdot (\frac{c'}{k})^k \cdot (1 - \frac{1}{r_{in}})^{m_i - k} \tag{11}$$

where $c' = 9er_{out}(1 - r_i)$ is a constant.

By plotting the distribution $P(Y_j)$, it is easy to find that the distribution of $Y_j$ concentrates around the neighborhood of $k = \frac{m_i}{a}$, and decreases significantly as $Y_j$ deviates from it.

# B  Proofs of Theoretical Results

## B.1  Proof of Theorem A.1

*Proof.* We pick arbitrary $0 < j, k \leq L$ and generates two sequences $\{\widehat{m}_i\}_{i=1}^L, \{\widetilde{m}_i\}_{i=1}^L$ as follows:

$$\begin{aligned} \widehat{m}_j = m_j - 1, \widehat{m}_k = m_k + 1, \widehat{m}_i = m_i \forall i \neq j, i \neq k. \\ \widetilde{m}_j = m_j + 1, \widetilde{m}_k = m_k - 1, \widetilde{m}_i = m_i \forall i \neq j, i \neq k. \end{aligned} \tag{12}$$

Consider $\{m_i\}_{i=1}^L$ the optimality that maximizes the combination number $\Pi_{i=1}^L \binom{n_i}{m_i}$. We have the following inequality:

$$\begin{aligned} 1 > \frac{\Pi_{i=1}^L \binom{n_i}{\widehat{m}_i}}{\Pi_{i=1}^L \binom{n_i}{m_i}} = \frac{m_j}{n_j - m_j + 1} \frac{n_k - m_k}{m_k + 1} \\ \\ 1 > \frac{\Pi_{i=1}^L \binom{n_i}{\widetilde{m}_i}}{\Pi_{i=1}^L \binom{n_i}{m_i}} = \frac{n_j - m_j}{m_j + 1} \frac{m_k}{n_k - m_k + 1} \end{aligned} \tag{13}$$

Reorganize the inequalities above, we obtain:

$$-\left(\frac{1}{n_k} + \frac{m_k - m_j + 1}{n_j n_k}\right) < \frac{m_k}{n_k} - \frac{m_j}{n_j} < \left(\frac{1}{n_j} + \frac{m_j - m_k + 1}{n_j n_k}\right) \tag{14}$$

Consider $1 \leq m_j \leq n_j$ and $1 \leq m_k \leq n_k$, we have $\frac{m_k - m_j + 1}{n_j n_k} \leq \frac{1}{n_j}$ and $\frac{m_j - m_k + 1}{n_j n_k} \leq \frac{1}{n_k}$. As a result, we have the following inequality:

$$\forall j, k, -\left(\frac{1}{n_j} + \frac{1}{n_k}\right) < \frac{m_k}{n_k} - \frac{m_j}{n_j} < \left(\frac{1}{n_j} + \frac{1}{n_k}\right) \tag{15}$$

This concludes the proof. $\square$

## B.2  Proof of Theorem A.2

*Proof.* We proof the theorem by contradictory. We assume the optimal $\{m_i\}_{i=1}^L$ does not satisfy the property mentioned in Theorem A.2. This means $\exists 1 \leq j \leq L$ such that $m_j < n_j$ and $\exists 1 \leq k \leq L, m_j < m_k - 1$. Based on this, we then construct a new sequence $\{\widehat{m}_i\}_{i=1}^L$ as follows:

$$\widehat{m}_j = m_j + 1; \widehat{m}_k = m_k - 1; \forall i \neq j, i \neq k, \widehat{m}_i = m_i. \tag{16}$$

We then calculate the ratio of $\Pi_{i=1}^L \widehat{m}_i$ and $\Pi_{i=1}^L m_i$:

$$\frac{\Pi_{i=1}^L \widehat{m}_i}{\Pi_{i=1}^L m_i} = \frac{(m_j + 1)(m_k - 1)}{m_j m_k} = 1 + \frac{m_k - m_j - 1}{m_j m_k} > 1 \tag{17}$$

The last inequality is based on the assumption $m_j < m_k - 1$. (17) indicates $\Pi_{i=1}^L \widehat{m}_i > \Pi_{i=1}^L m_i$, which contradicts the optimality of $\{m_i\}_{i=1}^L$. $\qquad\square$

## C  Algorithm

We provide the pseudo-code of the edge pop-up algorithm for adversarial robustness as Algorithm 1. We use PGD to generate adversarial attacks. $\Pi_{\mathcal{S}_\epsilon}$ mean projection into the set $\mathcal{S}_\epsilon$.

---

**Algorithm 1** Edge pop-up algorithm for adversarial robustness.

---

**Input:** training set $\mathcal{D}$, batch size $B$, PGD step size $\alpha$ and iteration number $n$, adversarial budget $\mathcal{S}_\epsilon$, pruning rate $r$, mask function $M$, the optimizer.
Random initialize the model parameters $\mathbf{w}$ and the scores $\mathbf{s}$.
**for** Sample a mini-batch $\{\mathbf{x}_i, y_i\}_{i=1}^B \sim \mathcal{D}$ **do**
  **for** i = 1, 2, ..., B **do**
    Sample a random noise $\delta$ within the adversarial budget $\mathcal{S}_\epsilon$.
    $\mathbf{x}_i^{(0)} = \mathbf{x}_i + \delta$
    **for** j = 1, 2, ..., n **do**
      $\mathbf{x}_i^{(j)} = \mathbf{x}_i^{(j-1)} + \alpha \nabla_{\mathbf{x}_i^{(j-1)}} \mathcal{L}(f(\mathbf{w} \odot M(\mathbf{s}, r), \mathbf{x}_i^{(j-1)}), y_i)$
      $\mathbf{x}_i^{(j)} = \mathbf{x}_i + \Pi_{\mathcal{S}_\epsilon}\left(\mathbf{x}_i^{(j)} - \mathbf{x}_i\right)$
    **end for**
  **end for**
  Calculate the gradient $\mathbf{g} = \frac{1}{B}\sum_{i=1}^B \nabla_{\mathbf{s}} \mathcal{L}(f(\mathbf{w} \odot M(\mathbf{s}, r), \mathbf{x}_i^{(n)}), y_i)$
  Update the score $\mathbf{s}$ using the optimizer.
**end for**
**Output:** the pruning mask $M(\mathbf{s}, r)$.

---

We provide the pseudo-code of our algorithm on the ImageNet100 as Algorithm 2. It incorporates FGSM [57] with ATTA [7]. In addition, due to the high resolution and large size of the ImageNet100 dataset, we need to compress the initial perturbation directory to reduce the overhead of memory consumption. Here, we choose to downsample the original perturbation to reduce its resolution for storage, and then upsample it back to the original resolution when using it as the initial perturbation.

## D  Experiments

### D.1  Experimental Settings

**General** The RN34 architecture we use in this paper is the same as the one in [51, 54], and it has 21265088 trainable parameters. The bias terms of all linear layers are initialized 0, and are thus disabled. We also disable the learnable affine parameters in batch normalization layers, following the setup of [51]. Unless specified, the number of training epochs for CIFAR10 and CIFAR100 is 400, and for ImageNet100 there are 100 training epochs. The adversarial budget in this paper is based on $l_\infty$ norm and the perturbation strength $\epsilon$ is $8/255$ for CIFAR10, $4/255$ for CIFAR100 and $2/255$ for ImageNet100. The resolution of CIFAR10 and CIFAR100 is $32 \times 32$; the resolution of ImageNet100 is $224 \times 224$. ImageNet100 is a subset of ImageNet which consists of 100 classes. The selection of these classes follows the settings of a python library called Continuum [20]. The PGD attacks used in our experiments have 10 iterations and the step size is one-quarter of the $\epsilon$, respectively. The AutoAttack (AA) consists of the following four attacks: 1) the untargeted 100-iteration AutoPGD based on cross-entropy loss; 2) the targeted 100-iteration AutoPGD based on difference of logits ratio (DLR) loss; 3) the targeted 100-iteration FAB attack [14]; 4) the black-box 5000-query Square

---

**Algorithm 2** Accelerated training for ImageNet100.

---

**Input:** training set $\mathcal{D}$, batch size $B$, FGSM step size $\alpha$, adversarial budget $\mathcal{S}_\epsilon$, pruning rate $r$, mask function $M$, the optimizer.

Random initialize the model parameters $\mathbf{w}$ and the scores $\mathbf{s}$.

Initialize the instance-to-perturbation dictionary $\mathcal{M} = \{\}$

**for** Sample a mini-batch $\{\mathbf{x}_i, y_i\}_{i=1}^{B} \sim \mathcal{D}$ **do**

    **for** $i = 1, 2, ..., n$ **do**

        Data augmentation $\mathbf{x}_i \leftarrow A(\mathbf{x}_i)$

        **if** $\mathbf{x}_i$ in $\mathcal{M}$ **then**

            Get the downsampled perturbation: $\delta_i' = A(\mathcal{M}(\mathbf{x}_i))$

            Upsample $\delta'$ to the original resolution and get $\delta_i$.

        **else**

            Sample a random noise $\delta_i$ within the adversarial budget $\mathcal{S}_\epsilon$

        **end if**

        $\delta_i \leftarrow \delta_i + \alpha \nabla_{\delta_i} \mathcal{L}(f(\mathbf{w} \odot M(\mathbf{s}, r), \mathbf{x}_i + \delta_i), y_i)$

        $\delta_i \leftarrow \Pi_{\mathcal{S}_\epsilon} \delta_i$

        Update the dictionary by the downsampled perturbation $\delta_i'$: $\mathcal{M}(\mathbf{x}_i) = A^{-1}(\delta_i')$

    **end for**

**end for**

Calculate the gradient $\mathbf{g} = \frac{1}{B} \sum_{i=1}^{B} \nabla_{\mathbf{s}} \mathcal{L}(f(\mathbf{w} \odot M(\mathbf{s}, r), \mathbf{x}_i + \delta_i), y_i)$

Update the score $\mathbf{s}$ using the optimizer.

**Output:** the pruning mask $M(\mathbf{s}, r)$.

---

attack [2]. We use the same hyper parameters in all these component attacks as in the original AutoAttack implementation.[6]

We train the model using an SGD optimizer, with the momentum factor being $0.9$ and the weight decay factor being $5 \times 10^{-4}$. The learning rate is initially $0.1$ and decays following the cosine annealing scheduler. Finally, since adversarial training suffers from severe overfitting [53], we use a validation set consisting of $2\%$ of the training data to select the best model during training.

**Adversarial Training** We apply the same settings as above to adversarial training, except the choice of optimizer and learning rate. For full precision networks, we use an SGD optimizer with an initial learning rate of $0.1$ and decreases by a factor of 10 in the 200th and 300th epoch for CIFAR10 and CIFAR100 models. For ImageNet100, the learning rate decreases by a factor of 10 in the 50th and 75th epoch. For binary networks, we use Adam optimizer [35] suggested in [10] and have a cosine annealing learning rate schedule with an initial learning rate of $1 \times 10^{-4}$.

**Baselines (FlyingBird(+), BCS, RST, HYDRA, ATMC)** Our results on the baselines are based on their original public implementation except that we use the validation set to pick the best model during training. FlyingBird(+), BCS, and HYDRA do not inherently support binary networks, so we plug in the *BinaryConnect* algorithm [10] with the same settings as the ones in adversarial training. We also plug in Algorithm 2 for fast training on ImageNet100. We scale down the number of training epochs of FlyingBird(+), BCS, RST to 100 epochs, and HYDRA to 110 epochs (50 pretrain + 10 prune + 50 finetune). For ATMC, we use 50 epochs for each of the four training phases, adding up to 200 epochs in total. In all baselines except RST, the batch normalization layers in the model have affine operations and are learnable. This introduces additional trainable parameters and is different from the network used in our method.

**Smaller RN34 Variants** Based on the adaptive pruning strategy, we designed several smaller RN34 variants with approximately the same number of parameters as the pruned networks. These variants have the same topology as RN34 but have fewer channels in each layer. In Table 7, we provide architecture details based on different values of $p$ when the pruning rate $r$ is $0.99$. The *Small RN34* model in Table 4 represents the small model with $p = 0.1$ (*Small RN34-p0.1* in Table 7), since it has better performance than the other small networks.

---

[6]AutoAttack: https://github.com/fra31/auto-attack.

| layer name | Small RN34-p0.1 | Small RN34-p1.0 |
|---|---|---|
| conv1 | $3 \times 3, 23$ | $3 \times 3, 6$ |
| Block1 | $\begin{bmatrix} 3 \times 3, 23 \\ 3 \times 3, 23 \end{bmatrix} \times 3$ | $\begin{bmatrix} 3 \times 3, 6 \\ 3 \times 3, 6 \end{bmatrix} \times 3$ |
| Block2 | $\begin{bmatrix} 3 \times 3, 25 \\ 3 \times 3, 25 \end{bmatrix} \times 4$ | $\begin{bmatrix} 3 \times 3, 13 \\ 3 \times 3, 13 \end{bmatrix} \times 4$ |
| Block3 | $\begin{bmatrix} 3 \times 3, 27 \\ 3 \times 3, 27 \end{bmatrix} \times 6$ | $\begin{bmatrix} 3 \times 3, 26 \\ 3 \times 3, 26 \end{bmatrix} \times 6$ |
| Block4 | $\begin{bmatrix} 3 \times 3, 29 \\ 3 \times 3, 29 \end{bmatrix} \times 3$ | $\begin{bmatrix} 3 \times 3, 51 \\ 3 \times 3, 51 \end{bmatrix} \times 3$ |
| | average pool, 10d-fc, softmax | |
| #params | 201078 | 216360 |

Table 7: RN34 variants that have similar layer sizes as the pruned RN34 obtained by different $p$ values. $3 \times 3 \times 23$ means the kernel size is $3 \times 3$ and there are 23 output channels.

## D.2 Additional Experimental Results

### D.2.1 Ablation Study in the Non-Adversarial Case

In the non-adversarial case, we train the models using clean inputs and report the clean accuracy in Table 8. Other hyper-parameters here are the same as in Table 3. Our conclusions from Table 3 also hold true here: the *binary initialization* can achieves comparable performance as the *Signed Kaiming Constant*; the last batch normalization layer helps improve performance for both initialization schemes.

| Prune Scheme | Signed KC | | Binary | |
| | no LBN | LBN | no LBN | LBN |
|---|---|---|---|---|
| $p = 0.0$ | 93.25 | 93.99 | 93.64 | **94.05** |
| $p = 0.1$ | 92.12 | 93.98 | **93.84** | 93.99 |
| $p = 0.2$ | 92.96 | **94.35** | 89.27 | 93.87 |
| $p = 0.5$ | **93.44** | 94.29 | 90.85 | 94.00 |
| $p = 0.8$ | 90.93 | 92.57 | 90.37 | 92.42 |
| $p = 0.9$ | 91.31 | 92.26 | 90.51 | 90.12 |
| $p = 1.0$ | 89.27 | 89.12 | 87.58 | 89.03 |

Table 8: The accuracy (in %) of vanilla trained models on the CIFAR10 test set under various settings, including *Signed Kaiming Constant* (Signed KC) and the binary initialization. We include models both with and without the last batch normalization layer (LBN). The best results are marked in bold.

### D.2.2 More results of Baselines

We show in Table 9 the complete set of experiments of baseline algorithms on CIFAR10 and CIFAR100 as a complementary of Table 4. Specifically, we compare baselines with different architectures and with different pruning strategies. First, the last batch normalization layer (LBN) does not improve the baselines that update model parameters in the full-precision setting, because the magnitude of the output logits can be automatically adjusted in these cases. There is no need to insert another normalization layer. For FlyingBird(+), BCS and HYDRA, adding LBN to a binary network will most likely be beneficial to a better performance. This observation is consistent with our claim in Appendix A.3. As for ATMC, it is actually not pruning a truly binary network since the value of model parameters are trainable and not necessarily $+1$ or $-1$, so adding LBN might not be useful in this case. For the pruning strategy, *adaptive pruning* strategy with $p = 0.1$ always has better performance than the *fixed pruning rate* strategy, i.e., $p = 1.0$. This is because the pruning rate here is very high $r = 0.99$, and we need a small value of $p$ based on the analysis in Section 3.2. Furthermore,

we provide the performance of TRADES [64], which trades clean accuracy for adversarial accuracy. Compared with adversarial training (AT), TRADES achieves competitive performance in the full precision cases, but it performance degrades significantly in the binary cases.

| Method | Architecture | Pruning Strategy | CIFAR10 | | CIFAR100 | |
|---|---|---|---|---|---|---|
| | | | FP | Binary | FP | Binary |
| AT | RN34 | Not Pruned | 43.26 | 40.34 | 36.63 | 26.49 |
| AT | RN34-LBN | Not Pruned | 42.39 | 39.58 | 35.15 | 32.98 |
| TRADES | RN34 | Not Pruned | 49.07 | 30.18 | 35.28 | 29.64 |
| TRADES | RN34-LBN | Not Pruned | 48.27 | 37.91 | 31.23 | 31.26 |
| FlyingBird | RN34 | Dynamic | 45.86 | 34.37 | 35.91 | 22.49 |
| FlyingBird+ | RN34 | Dynamic | 44.57 | 33.33 | 34.30 | 22.64 |
| FlyingBird | RN34-LBN | Dynamic | 45.58 | 37.18 | 35.06 | 24.94 |
| FlyingBird+ | RN34-LBN | Dynamic | 44.44 | 37.48 | 34.03 | 24.50 |
| BCS | RN34 | Dynamic | 43.51 | 22.61 | 31.85 | 11.96 |
| BCS | RN34-LBN | Dynamic | 42.02 | 30.67 | 31.16 | 17.97 |
| RST | RN34 | $p = 1.0$ | 34.95 | - | 21.96 | - |
| RST | RN34-LBN | $p = 1.0$ | 37.23 | - | 23.14 | - |
| HYDRA | RN34 | $p = 0.1$ | 42.73 | 29.28 | 33.00 | 23.60 |
| HYDRA | RN34 | $p = 1.0$ | 40.51 | 26.40 | 31.09 | 18.24 |
| HYDRA | RN34-LBN | $p = 0.1$ | 40.55 | 33.99 | 13.63 | 25.53 |
| HYDRA | RN34-LBN | $p = 1.0$ | 32.93 | 26.23 | 29.96 | 18.91 |
| ATMC | RN34 | Global | 34.14 | 25.62 | 25.10 | 11.09 |
| ATMC | RN34 | $p = 0.1$ | 34.58 | 24.65 | 25.37 | 11.04 |
| ATMC | RN34 | $p = 1.0$ | 30.50 | 20.21 | 22.28 | 2.53 |
| ATMC | RN34-LBN | Global | 33.55 | 19.01 | 23.16 | 15.73 |
| ATMC | RN34-LBN | $p = 0.1$ | 31.61 | 22.88 | 25.16 | 17.33 |
| ATMC | RN34-LBN | $p = 1.0$ | 27.88 | 13.22 | 22.12 | 9.55 |
| AT | Small RN34-p0.1 | Not Pruned | 42.01 | 32.54 | 28.46 | 16.18 |
| AT | Small RN34-p1.0 | Not Pruned | 38.81 | 26.03 | 27.68 | 15.85 |
| TRADES | Small RN34-p0.1 | Not Pruned | 42.60 | 29.92 | 28.44 | 15.25 |
| TRADES | Small RN34-p1.0 | Not Pruned | 38.53 | 24.83 | 27.63 | 13.16 |
| Ours | RN34-LBN | $p = 0.1$ | - | **45.06** | - | **34.83** |
| Ours | RN34-LBN | $p = 1.0$ | - | 34.57 | - | 26.32 |
| Ours (fast) | RN34-LBN | $p = 0.1$ | - | 40.77 | - | 34.45 |
| Ours (fast) | RN34-LBN | $p = 1.0$ | - | 29.68 | - | 24.97 |

Table 9: Robust accuracy (in %) on the CIFAR10 and CIFAR100 test sets for AT, TRADES, FlyingBird(+), BCS, RST, HYDRA, ATMC and our proposed method. "RN34-LBN" represents RN34 with the last batch normalization layer. "Small RN34" here refers to Small RN34-p0.1 in Table 7 of Appendix D.1. Among the compressed models, the best results for full precision (FP) models are underlined; the best results for binary models are marked in bold.

### D.2.3  Clean accuracy of Models in Table 4

Table 10 shows the accuracy on the clean test set of the models in Table 4. In the CIFAR10 dataset, our pruned networks with both normal and fast pruning achieve the highest vanilla accuracy among all binary networks. Although the accuracy is lower than full-precision networks by ATMC, our model performs notably better ($> 10\%$) under AutoAttack. In the CIFAR100 dataset, our model using FGSM with ATTA has the best vanilla accuracy among both full-precision networks and binary networks, and also achieves comparable robust accuracy to them, as shown in Table 4. Our model using PGD also achieves competitive performance, better than all other binary networks. In the ImageNet100 dataset, our model still outperforms all other pruned binary models, although it is worse than some full precision models. These results indicate that our models can achieve competitive robust accuracy without losing too much vanilla accuracy, hence more powerful in real applications where both robust and vanilla accuracy are important.

### D.2.4  Our Method in the Non-adversarial Cases

Vanilla training can be considered as a special case of adversarial training: the case when $\epsilon = 0$. Therefore, our methods, as well as baselines, are applicable to vanilla training. The results of the cases when $\epsilon = 0$ are demonstrated in Table 11. Since there are no adversarial attacks in vanilla training,

| Method | Architecture | Pruning Strategy | CIFAR10 FP | CIFAR10 Binary | CIFAR100 FP | CIFAR100 Binary | ImageNet100 FP | ImageNet100 Binary |
|---|---|---|---|---|---|---|---|---|
| AT | RN34 | Not Pruned | 80.99 | 74.37 | 61.48 | 47.87 | 78.98 | 63.76 |
| AT | RN34-LBN | Not Pruned | 80.96 | 74.17 | 57.73 | 60.08 | 77.66 | 64.60 |
| AT | Small RN34 | Not Pruned | 74.76 | 58.69 | 52.77 | 28.81 | 49.64 | 21.12 |
| FlyingBird | RN34 | Dynamic | 79.29 | 62.28 | $\underline{62.12}$ | 43.66 | 66.66 | 19.74 |
| FlyingBird+ | RN34 | Dynamic | 77.01 | 62.69 | 59.09 | 41.69 | 66.66 | 19.74 |
| BCS | RN34 | Dynamic | 74.75 | - | 53.82 | - | - | - |
| RST | RN34 | $p = 1.0$ | 65.93 | - | 38.87 | - | 42.70 | - |
| RST | RN34-LBN | $p = 1.0$ | 67.45 | - | 42.95 | - | 46.22 | - |
| HYDRA | RN34 | $p = 0.1$ | 75.31 | 62.09 | 55.92 | 45.96 | $\underline{67.76}$ | 33.18 |
| ATMC | RN34 | Global | $\underline{81.85}$ | 72.97 | 57.15 | 36.39 | 60.68 | 26.80 |
| ATMC | RN34 | $p = 0.1$ | 81.37 | 73.34 | 59.99 | 32.68 | 61.88 | 16.34 |
| Ours | RN34-LBN | $p = 0.1$ | - | 76.59 | - | 60.16 | - | 58.94 |
| Ours(fast) | RN34-LBN | $p = 0.1$ | - | **81.63** | - | **63.73** | - | |

Table 10: The accuracy (in %) on the clean inputs of the methods studied in Section 4.2. "RN34-LBN" represents RN34 with the last batch normalization layer. Among the pruned models, the best results in the full precision (FP) cases are underlined and the best results in the binary cases are marked in bold.

the acceleration used in "Ours (fast)" is not applicable here. The results in Table 11 demonstrate the consistent observations with Table 4: our proposed methods achieve the best performance among binary networks.

| Method | Architecture | Pruning Strategy | CIFAR10 FP | CIFAR10 Binary | CIFAR100 FP | CIFAR100 Binary | ImageNet100 FP | ImageNet100 Binary |
|---|---|---|---|---|---|---|---|---|
| AT | RN34 | Not Pruned | 94.80 | 90.11 | 76.39 | 70.02 | 80.26 | 68.26 |
| AT | RN34-LBN | Not Pruned | 94.79 | 92.46 | 76.85 | 73.49 | 79.84 | 73.88 |
| AT | Small RN34 | Not Pruned | 91.99 | 85.61 | 65.48 | 43.46 | 58.14 | 29.62 |
| FlyingBird | RN34 | Dynamic | $\underline{93.41}$ | 88.96 | 71.77 | 61.50 | 74.06 | 26.06 |
| FlyingBird+ | RN34 | Dynamic | 92.28 | 86.44 | $\underline{72.03}$ | 58.09 | 74.40 | 27.52 |
| BCS | RN34 | Dynamic | 90.69 | - | 67.39 | - | - | - |
| RST | RN34 | $p = 1.0$ | 88.43 | - | 56.65 | - | 50.18 | - |
| RST | RN34-LBN | $p = 1.0$ | 89.14 | - | 62.93 | - | 61.52 | - |
| HYDRA | RN34 | $p = 0.1$ | 91.13 | 88.10 | 68.84 | 62.10 | $\underline{76.42}$ | 49.40 |
| ATMC | RN34 | Global | 92.01 | 88.40 | 67.45 | 51.96 | 69.36 | 35.30 |
| ATMC | RN34 | $p = 0.1$ | 91.32 | 79.46 | 68.03 | 50.94 | 70.12 | 33.52 |
| Ours | RN34-LBN | $p = 0.1$ | - | **93.99** | - | **75.37** | - | **72.80** |

Table 11: Clean accuracy (in %) on the CIFAR10, CIFAR100 and ImageNet100 test sets for the baselines and our proposed method in the non-adversarial case, i.e., $\epsilon = 0$. "RN34-LBN" represents ResNet34 with the last batch normalization layer. "Small RN34" refers to Small RN34-p0.1 in Table 7 of Appendix D.1. The pruning rate is set to 0.99 except for the not-pruned methods. Among the pruned models, the best results for the full-precision (FP) models are underlined; the best results for the binary models are marked in bold.

### D.2.5 Mask of the Pruned Network

We have demonstrated that the masks of the pruned network obtained by our method are structured to some degree in Section 4.3. We have also analyzed the structure of a randomly pruned network in Appendix A.4.

Figure 5 shows the one of the convolutional layers in our pruned RN34 network. We resize the layer parameters as grids of shape $(r_{out}, r_{in})$ for visualization. Each grid represents a 3-by-3 kernel. So the shape of parameters is $(r_{out} \times 3, r_{in} \times 3)$. The retained parameters in each kernel are marked in blue. The pruning rate for this layer is $r = 0.99$. We highlight the input channels that are totally pruned in orange. We also use a white bar at the top of the figure to indicate these empty input channels.

In Section 4.3, we also point out the aligned pruning pattern in the two consecutive layers, layer1 and layer2, of the same residual block in RN34. Figure 3 shows their pruning masks. The side bars show which channel is non-empty(colored in blue). For convenience, layer1 is resized in $(r_{out} \times 3, r_{in} \times 3)$, and layer2 is organized in $(r_{in} \times 3, r_{out} \times 3)$. It is interesting that the pruned input channels of layer2 are well aligned with the pruned output channels of layer1.

Note that our finding also holds in the vanilla settings, i.e. pruning with clean examples. We think this observation enables a possible way for regular pruning.

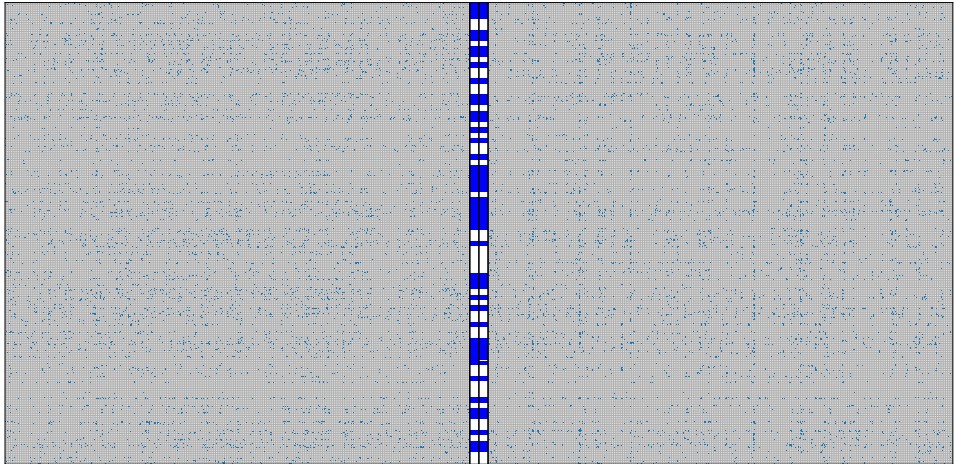

Figure 3: Distribution of weights in two consecutive layers. In layer1 (left), the masks are reshaped into $(r_{out} \times 3, r_{in} \times 3)$ while masks in layer2 (right) are reshaped into $(r_{in} \times 3, r_{out} \times 3)$. The output channels totally pruned in layer1 and the input channels totally pruned in layer2 are highlighted as the white bars in the middle. Due to the large number of parameters in these layers, readers could zoom in this figure to see more details.

### D.2.6 Learning Curves of Adaptive Pruning with Different $p$ Values

We plot the learning curves when we use the *adaptive pruning* strategy with different values of $p$ in Figure 4. Here, we use $r = 0.99$ and $r = 0.5$ as two examples. Based on the results of Table 1, our method achieves the best performance under $p = 0.1$ when $r = 0.99$ and under $p = 1.0$ when $r = 0.5$. The learning curves in Figure 4 indicate that the training process is quite unstable when using the inappropriate pruning strategy, leading to suboptimal performance.

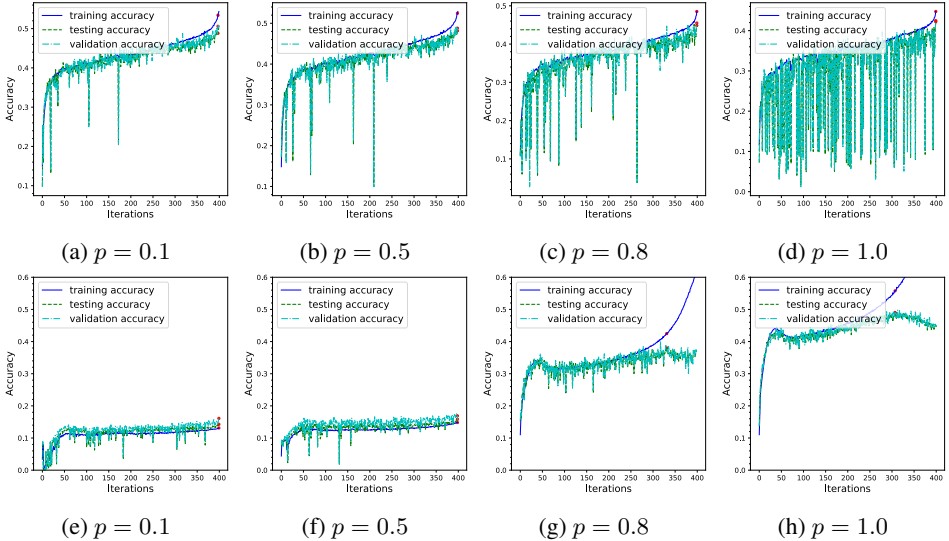

Figure 4: Learning curves of our proposed method under adaptive pruning strategy with different values of $p$. The pruning ratio is $0.99$ for figure (a) - (d) and is $0.5$ for figure (e) - (h).

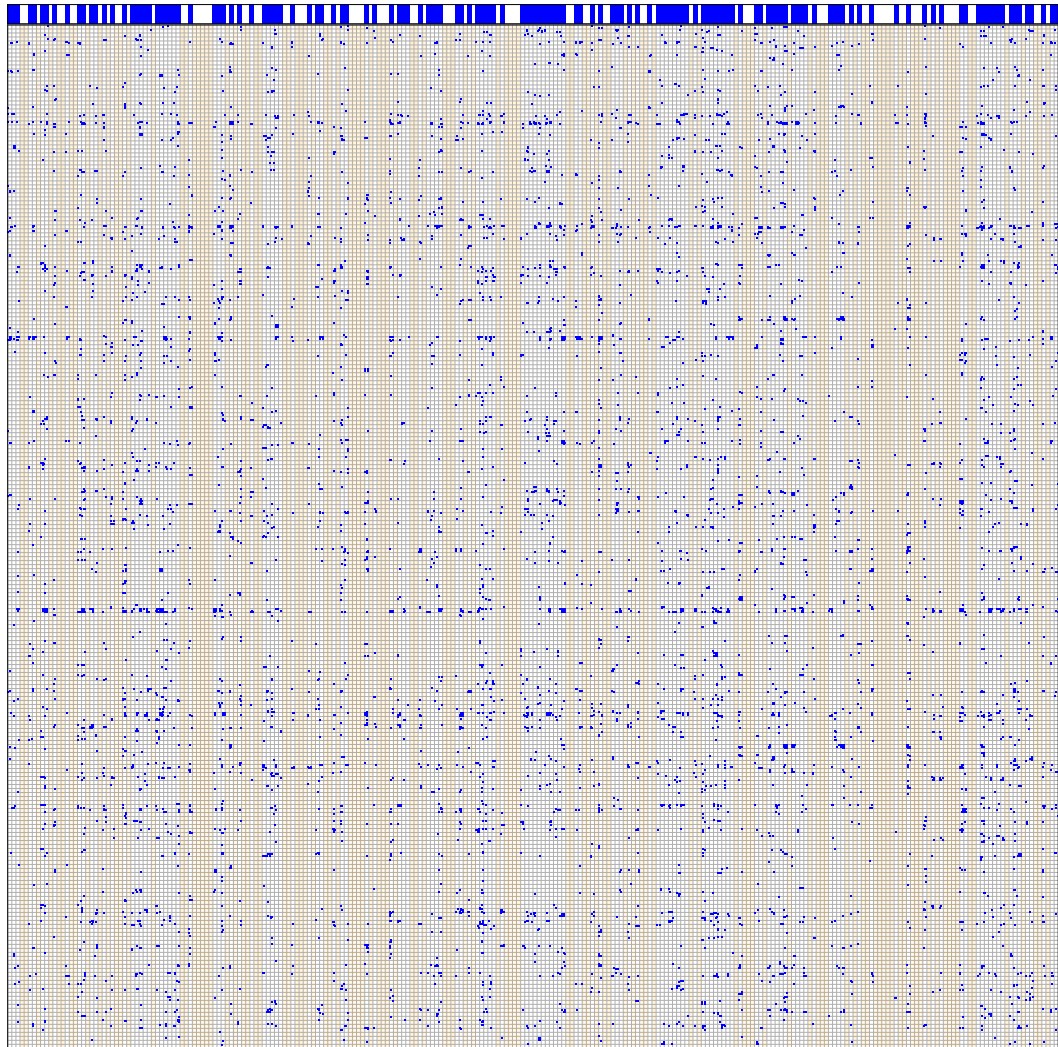

Figure 5: Mask visualization of the weight of a random convolutional layer in our model. The parameters retained is highlighted as blue dots. The dimension of the convolutional kernel is ($r_{out}$, $r_{in}$, 3, 3). We reshape this kernel in rectangle of shape ($r_{out} \times 3$, $r_{in} \times 3$). Channels with no remaining weight are colored orange. The top bar indicates whether the channel is empty (white) or not (blue). Due to the large number of parameters in this layer, readers could zoom in this figure to see more details.