# OpenReview forum: "Robust Binary Models by Pruning Randomly-initialized Networks"
_NeurIPS.cc/2022/Conference — NeurIPS 2022 Accept_

### Official Review · Reviewer_ATBX · 2022-07-10

**Rating:** 5
**Confidence:** 4
**Soundness:** 3 good
**Presentation:** 3 good
**Contribution:** 2 fair

**Summary:**

This manuscript extends the Strong Lottery Ticket Hypothesis to binary networks and proposed a new method to obtain compact binary networks with better robustness. The proposed method consists of three main parts: a) adaptively pruning different network layers; b) an effective binary initialization scheme; c) adding the last batch normalization layer. Extensive experiments on CIFAR-10/100 and ImageNet-100 validate the effectiveness of the method.

**Questions:**

None

**Limitations:**

To some extent, this paper may help to build more secure and compact deep learning systems in the real world. But the real-world system faces all types of attacks, which are far beyond adversarial examples.


**Strengths And Weaknesses:**

#### Strength:

1. The paper is well written with sufficient details of the proposed methods.
2. Experimental results on three datasets show superior performance against multiple baseline methods, including vanilla AT, Flying Bird, BCS, RST, HYDRA, and ATMC.
3. Analysis of the derived sparse patterns is provided.

#### Weakness:

1. The proposed methods search for sparse binary subnetworks within randomly-initialized networks. It's unclear whether such a scheme is better than training the parameters directly. It seems the proposed methods (adaptive pruning, binary initialization, and last batch normalization layer) can also to used to optimize the parameters of randomly-initialized networks.
2. As the proposed methods aim to obtain binary neural networks, which are both compact and robust. The comparison about training and inference cost (e.g., FLOPs) is important.
3. It seems better to firstly report the comparison results in Section 4.2, and then conduct the ablation study of three proposed methods (Section 4.1).

---

> ### Author Response · Authors · 2022-08-02
> **Response to Reviewer ATBX**
>
> We thank the reviewer for their constructive comments. Below are our point-to-point responses. We quote the reviewer’s comments in **bold**
>
> **It is unclear that whether such a scheme is better than training the parameters directly, It seems the proposed methods (adaptive pruning, binary initialization, and last batch normalization layer) can also to used to optimize the parameters of randomly-initialized networks.**
>
> This is the core idea of the Strong Lottery Ticket *Hypothesis*: By pruning the random network, we can obtain comparable performance to directly training the network. The major contribution of our paper is to extend the Strong Lottery Ticket Hypothesis to the case of robust binary networks.
>
> Furthermore, we don’t claim that the sparse binary models obtained by our method perform better than adversarial training on the unpruned original networks. Actually, adversarial training on the full network achieves better performance than our method on ImageNet100 (Table 10 in the appendix).  However, our method achieves better performance than the baselines that produce binary networks with roughly the same number of parameters. Binary networks with roughly the same number of parameters have similar training and inference complexity.
>
> Our approach can also be used to train the model parameters.  In Table 8 of the appendix, we incorporate adaptive pruning ($p = 0.1$) and last batch normalization layer (LBN) into baselines that optimize the model parameters, such as HYDRA and ATMC. Their performance is still lower than our method.
>
> **The comparison about the training and inference cost (e.g. FLOPs) is important**
>
> In Section A.2 of the appendix, we discuss the number of FLOPs for full precision and binary networks in detail, both for their forward passes and their backward passes. The results are summarized in Table 5 of the appendix. For the ResNet34 model we use in most experiments, its binary version saves 45% and 32% FLOPs compared with its full-precision version in the training phase and inference phase, respectively.
>
> **It seems better to firstly report the comparison results in Section 4.2, and then conduct the ablation study**
>
> To avoid confusion about which table we refer to in our response, we keep the table numbering the same and thus do not adjust the order of the sections. We will do this in the future.
>
> **But the real-world system faces all types of attacks, which are far beyond adversarial examples.**
>
> We agree with the reviewer about the limitations of the norm-based adversarial budget, but this is not the focus of this paper. How to enrich the types of adversarial examples is still an open question of the community. However, we need to point out that adversarial robustness against norm-based attacks is also a very strong baseline for robustness against other common corruptions, as indicated in [A].
>
> [A]. K Kireev*, M Andriushchenko*, N Flammarion. On the effectiveness of adversarial training against common corruptions. UAI 2022.

---

> > ### Comment · Reviewer_ATBX · 2022-08-07
> > **Thanks for the responses**
> >
> > I would thank the authors for their efforts in the rebuttal. The responses address my concern and I raise my score to 5.

---

> > > ### Author Response · Authors · 2022-08-07
> > > **Thank You**
> > >
> > > Dear Reviewer ATBX,
> > >
> > > Thank you for your response and we are happy that our responses have addressed your concerns. If you have further suggestions for us to polish up the paper, please let us know.
> > >
> > > Paper 6197 Authors.

---

### Official Review · Reviewer_HVEC · 2022-07-11

**Rating:** 7
**Confidence:** 3
**Soundness:** 2 fair
**Presentation:** 3 good
**Contribution:** 2 fair

**Summary:**

This paper presents a method for searching binary randomly initialized networks to generate pruned and robust models. The paper leverages the existing edge-popup algorithm for pruning untrained networks via learning important scores instead of model weights. The paper also proposes adding a last layer batchnorm to stabilize the training of their compressed robust models. The paper conducts several experiments to support their claim of achieving state-of-the-art robust binary networks.

**Questions:**

1) In the Signed Kaiming Constant initialization (SKC) in line 175, is the 1-r a typo? since the original paper does not have the pruning ratio in the initialization step. If it is not a typo, does that mean every pruning ratio requires a different randomly initialized network?

2) Is the last normalization layer a standard batchnorm layer, or a homogenous (affine=False) batchnorm layer as used in other layers?

3) The observation that more pruning leads to higher robustness is very intriguing, especially when existing pruning + adversarial training methods, such as HYDRA [1] and ADMM [2], typically show that robustness decreases with the pruning rate (as is expected). Moreover, the edge-pop up algorithm typically achieves its best performance (in the non adversarial setting) when r=0.5, as it maximizes the number of possible network choices. Could the authors shed more light on the reason why their method seems to contradict both these existing observations?

4) If I understood correctly, the proposed method is able to generate 99% pruned binary networks, that achieve adversarial robustness that is higher than all other baselines, which includes full precision adversarially trained models and their pruned versions, as well as other trained binary models. For context, the state-of-the-art 99% pruned full precision baseline achieved by HYDRA suffers massive degradation in terms of robustness, when compared to an unpruned baseline. The results reported in this paper are very impressive and also very intriguing, as they seems too good to be true. I believe more experiments/analysis is warranted to explain why this method is so much more effective than prior work. One simple experiment perhaps is to initialize the weights to constant values between [-1,1] instead of the binary {-1,1}, and keep everything else the same. One would expect that pruning this baseline should have even better robust accuracy than the binarized version, and thus surpass that of any FP baseline. Another question is what would happen if we train the pruned models from scratch (that is take the pruning mask and initialize another model for adversarial training). It would be great if the authors can address this.

[1] Sehwag, V., Wang, S., Mittal, P., & Jana, S. (2020). Hydra: Pruning adversarially robust neural networks. Advances in Neural Information Processing Systems, 33, 19655-19666.

[2] Ye, S., Xu, K., Liu, S., Cheng, H., Lambrechts, J. H., Zhang, H., ... & Lin, X. (2019). Adversarial robustness vs. model compression, or both?. In Proceedings of the IEEE/CVF International Conference on Computer Vision (pp. 111-120).

**Limitations:**

The authors claim that their work is generic and does not suffer from limitations or potential negative societal impact.

**Strengths And Weaknesses:**

Strengths:
- The paper is well motivated
- The quality of writing is good, the ideas are well explained
- The experimental results support the authors' claims, achieving state-of-the-art results for pruned binary robust models

Weaknesses:
- The novelty of this work is a bit limited, as it combines existing techniques (edge-popup + adversarial loss similar to [1] and [2]), that being said the end results are very impressive
- I would have liked to see more analysis on why the method is so successful, as opposed to adversarial training and/or binarization/pruning schemes


[1] Ramanujan, V., Wortsman, M., Kembhavi, A., Farhadi, A., & Rastegari, M. (2020). What's hidden in a randomly weighted neural network?. In Proceedings of the IEEE/CVF Conference on Computer Vision and Pattern Recognition (pp. 11893-11902).

[2] Sehwag, V., Wang, S., Mittal, P., & Jana, S. (2020). Hydra: Pruning adversarially robust neural networks. Advances in Neural Information Processing Systems, 33, 19655-19666.

---

> ### Author Response · Authors · 2022-08-02
> **Response to Reviewer HVEC (part 1)**
>
> We thank the reviewer for their constructive comments. Below are our point-to-point responses. We quote the reviewer’s comments in **bold**
>
> **I would have liked to see more analysis on why the method is so successful, as opposed to adversarial training and/or binarization/pruning schemes**
>
> As discussed in Section 3.2, adaptive pruning is important for performance improvement when the pruning rate $r$ is close to $1$. This is because we should retain a higher proportion of parameters for small layers in this case. The last batch normalization layer (LBN) is important because it mitigates gradient vanishing and explosion as discussed in Section A.3 of the appendix.
>
> Most of the baselines focus on pruning full-precision networks, and they achieve competitive performance in this case. However, these methods are not specifically designed for binary networks, so they cannot handle the accuracy degradation in the binary case. Furthermore, our method focuses on learning the mask of the subnetwork instead of training the model parameters, whereas some baselines (such as HYDRA) also have pre-training and fine-tuning phases. Under the same training budget, these baselines have fewer epochs for learning the pruning masks. This may lead to sub-optimality.
>
> We do not claim that our pruned subnetworks always outperform the unpruned ones obtained by adversarial training. In fact, our pruned subnetworks perform worse than the adversarially trained unpruned models on ImageNet100 (as shown in Table10 in the appendix). However, our sparse networks always perform better than the small but dense networks with approximately the same number of parameters (Small RN34-p0.1 and Small RN34-p1.0 in Table 8 of the appendix) trained by adversarial training.
>
> **In the Signed Kaiming Constant initialization (SKC) in line 175, is the 1-r a typo? since the original paper does not have the pruning ratio in the initialization step. If it is not a typo, does that mean every pruning ratio requires a different randomly initialized network?**
>
> No, this is not a typo. In the original paper [A], the authors mention the scaling factor in Section 4.5 and explain the rationale behind it. Therefore, we apply the scaling factor to SKC in our experiments. Based on this, different pruning ratios $r$ mean different variances of the random initial parameters for SKC initialization.
>
>
> By contrast, in our binary network, we do not require such a scaling factor; all the weights in the binary networks are uniformly sampled from {-1, 1}.
>
> [A] V Ramanujan，What's hidden in a randomly weighted neural network?， CVPR 2020.
>
> **Is the last normalization layer a standard batchnorm layer, or a homogenous (affine=False) batchnorm layer as used in other layers?**
>
> Yes, it is a homogenous batchnorm layer, same as the other batchnorm layers used in our method.
>
> **The observation that more pruning leads to higher robustness is very intriguing, especially when existing pruning + adversarial training methods, such as HYDRA [1] and ADMM [2], typically show that robustness decreases with the pruning rate (as is expected). Edge-pop up algorithm typically achieves its best performance (in the non adversarial setting) when r=0.5, as it maximizes the number of possible network choices** // **Could the authors shed more light on the reason why their method seems to contradict both these existing observations?**
>
> We do not claim that more pruning leads to better robustness. In our experiments, we observed that $r=0.99$ is the optimal pruning rate, as shown in Table1. When the pruning rate becomes higher, such as $r = 0.995$ and $r = 0.998$, the performance also decreases.
>
> Methods such as HYDRA and ADMM update the model parameters, while our method fixes them and only updates the mask. In this case, our method should demonstrate a similar performance trend as Edge-popup instead of HYDRA or ADMM. For HYDRA and ADMM, the performance decreases with the increase of $r$; for Edge-popup and our method, the performance first increases and then decreases with the increase of $r$.
>
> Furthermore, the adaptive pruning strategy affects the optimal value of $r$. When we apply the same pruning rate in all layers, i.e., p=1.0, we observe a similar optimal value of $r$ as Edge-popup based on the results in Table1. Moreover, according to [B], adversarial robustness encourages a lower Lipschitz constant of the model, since higher model sparsity usually leads to a lower Lipschitz constant. Therefore, it is expected to see that the best performance is achieved with a larger pruning rate (0.99 in our experiments in Table 1).
>
> [B] Y. Guo, C. Zhang, C. Zhang, and Y. Chen, “Sparse DNNs with Improved Adversarial Robustness,” in Advances in Neural Information Processing Systems, 2018, vol. 31.

---

> > ### Author Response · Authors · 2022-08-02
> > **Response to Reviewer HVEC (part 2)**
> >
> > **If I understood correctly, the proposed method is able to generate 99% pruned binary networks, that achieve adversarial robustness that is higher than all other baselines, which includes full precision adversarially trained models and their pruned versions, as well as other trained binary models.**
> >
> > Not necessary. For example, in our experiments on ImageNet100 in Table 10 in Appendix D.2.4, our pruned subnetwork does not perform better than unpruned models obtained by adversarial training, both in the full-precision and binary cases. We do not claim that our method is better than adversarial training with unpruned networks. However, our method achieves the best performance among the binary networks with approximately the same number of parameters. That is to say, we achieve the best performance among models of approximately the same complexity.
> >
> > **One simple experiment perhaps is to initialize the weights to constant values between [-1,1] instead of the binary {-1,1}, and keep everything else the same. One would expect that pruning this baseline should have even better robust accuracy than the binarized version, and thus surpass that of any FP baseline.**
> >
> > We conduct experiments by changing the initialization scheme to uniform values [-1, 1]. We see performance degradation compared with the binary initialization: the robust test accuracy decreases from 45.06% for binary initialization to 41.47% for uniform initialization. Therefore, we don’t think initializing the weight as some continuous values can improve the performance. This is because we are not updating the model weights at all during the process, the continuous value of the initialization is not necessarily better than the discrete value {-1, +1}.
> >
> > Our observation is somewhat consistent with the vanilla full-precision cases in [48] (what’s hidden in a randomly weighted neural network. CVPR), which shows a performance degradation from the discrete Signed Kaiming Constant initialization to the continuous Kaiming Normal initialization.
> >
> >
> > **Another question is what would happen if we train the pruned models from scratch (that is take the pruning mask and initialize another model for adversarial training). It would be great if the authors can address this.**
> >
> > It is a very interesting question. We show below the experiments of retraining and fine-tuning from the pruned ResNet34 structure. All the results are the robust test accuracy based on AutoAttack in percentage. Fine-tuning from the pruned subnetwork improves the performance when we don’t apply LBN, but it only brings little improvement or even harms the performance with LBN added. Retraining the subnetwork does not yield a better performance regardless of LBN.
> >
> > | Dataset | LBN? | Pruned | Retrained | Finetuned |
> > | ------- | ----  | ------  | ---------  | ---------- |
> > | Cifar10 | Yes     | 45.06   | 40.18        | 44.31        |
> > | Cifar10 | No      | 41.01   | 40.37        | 43.22        |
> > | Cifar100 | Yes   | 34.83   | 28.04       |  35.39        |
> > | Cifar100 | No    | 28.71   | 27.84        | 31.01        |

---

> > > ### Comment · Reviewer_HVEC · 2022-08-08
> > > **Response to the authors**
> > >
> > > I thank the authors for their detailed response. Most of my questions are addressed. I will raise my score from 6 to 7.

---

> > > > ### Author Response · Authors · 2022-08-08
> > > > **Thank you**
> > > >
> > > > Dear Reviewer HVEC,
> > > >
> > > > Thank you for your response and your efforts to review our paper. We are happy that our responses have addressed most of your concerns. Please let us know if you have further comments or suggestions for us to improve our paper.
> > > >
> > > > Paper 6197 Authors

---

### Official Review · Reviewer_WydW · 2022-07-11

**Rating:** 4
**Confidence:** 4
**Soundness:** 3 good
**Presentation:** 3 good
**Contribution:** 3 good

**Summary:**

For a robust binary model built with Strong Lottery Ticket Hypothesis, the paper proposes a model compress method with three strategies: adaptive layer-wise pruning, a binary initialization scheme, and adding a last batch normalization layer. Experiments on CIFAR-10 and CIFAR-100 show that the proposed method outperforms previous competitive methods such as ATMC [22].

**Questions:**

1) Are the proposed three techniques useful for regular DNNs in addition to the robust binary models [19]?
2) Why do the methods in Table 4 use different $p$ value? Are the methods in Table 4 fairly compared with each other?
3) Is the proposed method scalable to large-scale datasets such as tinyImageNet?

**Limitations:**

As mentioned above, a major limitation is that the paper's proposed three techniques are not systematical but auxiliary model compression strategies.

**Strengths And Weaknesses:**

Strengths.
1) In Section 1 and Section 3.1 (Equation (2)), the paper clearly introduces the base model [19] that the paper works on: a robust binary model built with Strong Lottery Ticket Hypothesis.
2) In Section 3.2 and Section 3.2, introduction of the proposed three techniques is easy-to-follow, except too more "In Appendix ...". The proposed three techniques, adaptive layer-wise pruning, a binary initialization scheme, and adding a last batch normalization layer, are reasonable.
3) Experiments on CIFAR-10 and CIFAR-100 show that the proposed method outperforms previous competitive methods such as ATMC [22]. The paper further provides necessary analysis on the subnetwork patterns, and sensitivity analysis on the pruning strategy and added last batch-normalization layer.

Weaknesses.
1) My major concern is that the paper's proposed three techniques are not systematical but auxiliary model compression strategies. For example, the proposed adaptive pruning is a straightforward trade-off between fixed pruning rate and fixed number of parameters, and the key parameter $p$ is manually but not automatically selected in all experiments. Moreover, the paper works on a base model [19] which is already a robust binary model. The paper focuses on model compression only but not further improving binarization and adversarial robustness.
2) In Table 4, different methods use different $p$, e.g., $p=1.0$ for RST and $p=0.1$ for the proposed method. I think all methods should use the same $p$ value for fair comparison.
3) Since tiny-scale datasets CIFAR-10 and CIFAR-100 are easily to be overfitting. I suggest the authors use large-scale datasets such as tinyImageNet for experiments.

---

> ### Author Response · Authors · 2022-08-02
> **Response to Reviewer WydW**
>
> We thank the reviewer for their constructive comments. Below are the point-to-point responses. We quote the reviewer’s comments in **bold**
>
> **the key parameter p is manually but not automatically selected in all experiments**
>
> Table 1 demonstrates the correlation between $r$ and $p$. Furthermore, we use the same value of $p$ when $r$ is fixed to $0.99$. This facilitates the selection of $p$ for practitioners. That is, we can use the same value of $p$ when we use different architectures (ResNet34 in Table 4, ResNet18 and ResNet50 in Table 9 of the appendix), different datasets (CIFAR10 and CIFAR100 in Table 4, and ImageNet100 in Table 10) and different algorithms ($p = 0.1$ achieves better performance than $p = 1$ for all baselines in the ablation study in Table 8 of the appendix.)
>
> **Moreover, the paper works on a base model [19] which is already a robust binary model.**
>
> The work [19] does not study the case of binary models; the outcome of its algorithm is a full-precision model. By contrast, our models are binary and thus more efficient. Furthermore, techniques such as adaptive pruning and last batch normalization layer improve the performance of our binary subnetwork. In terms of robustness, our method consistently performs better than the one in [19].
>
> **In Table 4, different methods use different p, e.g., p=1.0 for RST and p=0.1 for the proposed method. I think all methods should use the same p value for fair comparison.** // **Why do the methods in Table 4 use different p values? Are the methods in Table 4 fairly compared with each other?**
>
> In addition to Table 4, more results about our proposed methods and baselines under different settings are provided in Table 3 in the experiment section and Table 8 in the appendix.
>
> RST can be considered as the base of our method, without adaptive pruning, last batch normalization layer, nor binary initialization. We have provided an ablation study of these three techniques in Table 3, which shows that adaptive pruning and last batch normalization layer improve the performance, and that binary initialization improves the efficiency with comparable performance.
>
> For other baselines, we provide additional results with different configurations, including $p = 0.1$ for baselines if applicable, in Table 8 of the appendix. The results in Table 8 still show that our method yields the best performance in robust accuracy among all binary networks.
>
> **I suggest the authors use large-scale datasets such as tinyImageNet for experiments** // **Is the proposed method scalable to large-scale datasets such as tinyImageNet?**
>
> In Table 10 of the appendix, we compare our method with all the baselines on the ImageNet100 dataset. ImageNet100 is a larger dataset containing images of the same resolution as ImageNet (224x224). However, TinyImageNet has a much lower resolution (64x64) than ImageNet. As shown in [A], higher-solution images are more sensitive to adversarial attacks. Moreover, ImageNet100 has more training instances than TinyImageNet. Therefore, we can say that ImageNet100 is a dataset of larger scale than TinyImageNet.
>
> The results on ImageNet100 in Table 10 of the appendix are consistent with the ones on CIFAR10 and CIFAR100: Our method achieves the best performance among all the binary networks.
>
> [A] C-J Simon-Gabriel et al. First-Order Adversarial Vulnerability of Neural Networks and Input Dimension. ICML 2019.
>
>
> **Are the proposed three techniques useful for regular DNNs in addition to the robust binary models [19]?**
>
> Vanilla training can be considered as a special case of adversarial training, where $\epsilon = 0$. Not only our proposed method but also all the baselines are applicable to vanilla training, through which we can obtain regular DNNs. However, adversarial training with $\epsilon > 0$ is more challenging, and thus we have to use a more flexible and powerful method to find the robust sub-networks. That is to say, the techniques proposed in this work are motivated by the cases of adversarial training. The results in our paper indicate that our method yields larger improvements in the robust DNN case, compared with the regular DNN one.
>
> The results of our method applied in the case where $\epsilon = 0$ are shown in Table 7 of the appendix (vanilla settings for ablation studies in Table 3). The performance of the baselines in Table 4 in vanilla training is shown in Table 12 in the appendix of our revised manuscript.

---

> > ### Comment · Reviewer_WydW · 2022-08-06
> > **The authors have addressed my concerns**
> >
> > Thanks for the authors' response. The authors have addressed my concerns.

---

> > > ### Author Response · Authors · 2022-08-06
> > > **Thank you and Additional Feedback**
> > >
> > > Dear Reviewer WydW,
> > >
> > > We are happy to hear that our responses have addressed your concerns. Could you please kindly consider adjusting your rating? Or do you have additional suggestions for us to improve our paper? We are happy to discuss this with you if any.
> > >
> > > Thank you.
> > >
> > > Paper6197 Authors

---

> > > ### Author Response · Authors · 2022-08-08
> > > **Waiting for the reviewer‘s response**
> > >
> > > Dear Reviewer WydW,
> > >
> > > The author-reviewer discussion period will finish soon, we notice your rating is a borderline reject. In addition to your original comments we have addressed in our rebuttal, do you have additional concerns or comments?
> > >
> > > We really appreciate your time and effort to review our paper.
> > >
> > > Paper6197 Author

---

### Official Review · Reviewer_WyYF · 2022-07-12

**Rating:** 7
**Confidence:** 3
**Soundness:** 4 excellent
**Presentation:** 4 excellent
**Contribution:** 3 good

**Summary:**

The paper proposes a method for learning robust binary neural networks from random initialization. The proposed training scheme includes  adaptive pruning approach and an edge pop-up algorithm for training the binary model to obtain a binary mask for pruning using a PGD-like method. It shows clear improvement over other adversarial training methods for binary models in experiments. Some theoretical analysis for the adaptive pruning method is provided.

**Questions:**

- Can the authors provide some intuition or analysis regarding why this training scheme could work?

**Limitations:**

- It would be interesting to investigate whether the proposed training method can also be applied to training binary Transformers.

**Strengths And Weaknesses:**

### Strengths
- The training algorithm is clearly stated makes sense.
- The experimental results are impressive with evident improvement over comparing methods.
- Some theoretical analysis for the adaptive pruning method is provided.
- The ablation study and sensitivity analysis are very clear and are consistent with intuition. For example, the ablation study for the hyper-parameter $p$ for adaptive pruning aligns very well with the analysis.

### Weakness
- Only one neural net model (ResNet34) is evaluated in the experiments.

---

> ### Author Response · Authors · 2022-08-02
> **Response to Reviewer WyYF**
>
> We thank the reviewer for their constructive comments. Below are our point-to-point responses. We quote the reviewer’s comments in **bold**
>
> **Only one neural net model (ResNet34) is evaluated in the experiments**
>
> In Table 9 of the appendix, we conduct experiments using ResNet18 and ResNet50. The results are consistent with the ResNet34 case in Table 4. This demonstrates that our method outperforms the baselines on other architectures as well.
>
> **Can the authors provide some intuition or analysis regarding why this training scheme could work?**
>
> In Section 3.2, we demonstrate the advantages of *adaptive pruning*; it is a flexible framework that assigns a higher proportion of unpruned parameters to small layers. It is also a trade-off between the fixed pruning rate strategy and the fixed number of parameter strategy. These are two extreme strategies with degraded performance for different pruning rates $r$; the fixed pruning rate strategy performs poorly when $r$ is close to $1$. By contrast, the fixed number of parameters strategy performs poorly with small $r$.
>
> The advantage of the last batch normalization layer (LBN) is briefly discussed in Section 3.3 and theoretically justified in the appendix (Section A.3) in detail. The LBN stabilizes the gradients for binary initialization. Without LBN, one can expect gradient vanishing or explosion.
>
> The advantage of binary initialization is mainly efficiency. We compare the computational complexity in FLOPs of full-precision weights and binary weights in the appendix (Section A.2). Binary parameters save approximately 45% and 32% FLOPs for the ResNet34 model we use in the experiments.
>
> **It would be interesting to investigate whether the proposed training method can also be applied to training binary transformers**
>
> As far as we know, pretraining on a larger dataset is necessary for vision transformers [A]. However, one of the main goals of this paper is efficiency. Pretraining binary transformers would incur a huge computational overhead, and thus we do not use transformers as the model architecture.
>
> [A] A. Dosovitskiy et al. An Image is Worth 16x16 Words: Transformers for Image Recognition at Scale. ICLR 2021.

---

> > ### Author Response · Authors · 2022-08-08
> > **Waiting for the reviewer's feedback**
> >
> > Dear Reviewer WyYF,
> >
> > The author-reviewer discussion period will end soon, we are looking forward to your feedback to our response.
> > Do we well address your concerns about our paper?
> >
> > We really appreciate your time and effort to review our paper.
> >
> > Paper6197 Authors

---

### Author Response · Authors · 2022-08-02
**General Response and Revision**

General Response

We thank the reviewers for their valuable feedback. In addition to the point-to-point responses to each reviewer, we made the following major revisions to our paper:
1. We add a new section in Appendix D.2.6 to show the accuracy (in %) on the CIFAR10 and CIFAR100 test sets for the baselines and our proposed method in vanilla training, i.e., $\epsilon$ = 0. The results are consistent with the adversarial cases.

Note that some of the reviewers’ concerns are addressed in the appendix of our original submission:
1. Section A.2 analyzes the number of FLOPs of the full-precision and binary networks; the results are summarized in Table 5.
2. Section A.3 theoretically justifies why the last batch normalization layer can mitigate gradient vanishing and gradient explosion.
3. The complete set of baseline experiments are provided in Table 8, including baselines using adaptive pruning and the last batch normalization layer if applicable. Our method still performs the best among all the binary networks.
5. Experiments using other model architectures, including ResNet18 and ResNet50, are provided in Table 9.
6. Experiments on ImageNet100, a much larger dataset consisting of higher resolution images, are summarized in Table 10.

---

### Meta-Review · Area_Chair_gB7P · 2022-08-31

**Recommendation:** Accept
**Confidence:** Less certain

**Metareview:**

The reviewers agree the paper studies an interesting problem on training robust binary neural networks and the paper does a good job in evaluating the proposed approach on multiple datasets and compares well with baselines. However the paper also has some drawbacks such as the proposed method has limited novelty and is a combination of existing techniques, missing comparisons to standard training based approaches, evaluations limited to ResNets.  Overall the paper is on borderline. I suggest acceptance and encourage the authors to include all the changes that came up during discussion in the final version and discuss the limitations.

**Award:**

No

---

### Decision · Program_Chairs · 2022-09-14

Accept